# Insights in a remote cryosphere: A multi method approach to assess permafrost occurrence at the Qugaqie basin, western Nyainqêntanglha Range, Tibetan Plateau

Johannes Buckel[1], Eike Reinosch[2], Andreas Hördt[1], Fan Zhang[3], Björn Riedel[2], Markus Gerke[2], Antje Schwalb[4], Roland Mäusbacher[5]

[1]Institute for Geophysics and extraterrestrial Physics, Technische Universiät Braunschweig, Braunschweig, 38106, Germany
[2]Institute for Geodesy and Photogrammetry, Technische Universiät Braunschweig, Braunschweig, 38106, Germany
[3]Key Laboratory of Tibetan Environment Changes and Land Surface Processes, Institute of Tibetan Plateau Research, Chinese Academy of Sciences, Beijing, 100101, China.
[4]Institute of Geosystems and Bioindication, Technische Universiät Braunschweig, Braunschweig, 38106, Germany
[5]Geographical Institute, Friedrich Schiller University of Jena, Jena, 07743, Germany

*Correspondence to*: Johannes Buckel (j.buckel@tu-braunschweig.de)

**Abstract.** Permafrost as a climate-sensitive parameter, its occurrence and distribution plays an important role in the observation of global warming. However, field-based permafrost distribution data and information on the subsurface ice content in the large area of the southern mountainous Tibetan Plateau (TP) is very sparse. Existing models based on boreholes and remote sensing approaches suggest permafrost probabilities for most of the Tibetan mountain ranges. Field data to validate permafrost models are generally lacking because access of the mountain regions in extreme altitudes is limited. The study provides geomorphological and geophysical field data from a north-orientated high-altitude catchment in the western Nyainqêntanglha Range. A multi-method-approach combines (A) geomorphological mapping, (B) electrical resistivity tomography (ERT) to identify subsurface ice-occurrence, and (C) Interferometric Synthetic Aperture Radar (InSAR) analysis to derive multi-annual creeping rates. The combination of the resulting data allows an assessment of the lower occurrence of permafrost in a range of 5350 and 5500 m above sea level (a.s.l.) in the Qugaqie basin. Periglacial landforms such as rockglaciers and protalus ramparts are located in the periglacial zone from 5300 – 5600 a.s.l. The altitudinal periglacial landform distribution is supported by ERT data detecting ice-rich permafrost in a rockglacier at 5500 m a.s.l. and ice lenses around the rockglacier (5450 m a.s.l.). The highest, multiannual creeping rates up to 150 mm/y are observed typically on these rockglaciers. This study closes the gap of unknown state of periglacial features and potential permafrost occurrence in a high-elevated basin at the western Nyainqêntanglha Range (Tibetan Plateau).

## 1 Introduction

Information on permafrost (defined as a thermal state of perennially cryotic ground, at least frozen of two consecutive years
(Ballantyne, 2018; Washburn, 1979)) distribution is of great importance in times of global warming, especially in high
mountain areas (Hock et al., 2019), because these areas are climatically sensitive (Barsch, 1996; Mollaret et al., 2019). The
International Panel on Climate Change (IPCC) reported for 2019 the strongest observed increase of permafrost temperature
(globally averaged across polar and high-mountain regions) since 2007 (Hock et al., 2019). Periglacial landforms, like
rockglaciers and protalus ramparts in this study, are features "resulting from the action of intense frost, often combined with
the presence of permafrost" (French, 2012). If permafrost as perennial frozen ground ice is available, periglacial landforms
are particularly well suited to detect and to study changes of permafrost and the related ice content (Kneisel and Kääb, 2007,
Kääb, 2013, Knight et al., 2019). These changes have an increasing impact on people and their livelihood (Gruber et al.,
2017), e.g., due to the importance of long-term ground ice as water resource (Jones et al., 2019) in arid/semiarid regions like
the Andes (Azócar and Brenning, 2010; Rangecroft et al., 2016) or the Tian Shan (Bolch and Gorbunov, 2014; Bolch and
Marchenko, 2006). The frozen water storages have a strong impact on water budgets by permafrost degradation and glacier
melt (Bibi et al., 2018; Song et al., 2020), especially at the so-called Asian water tower, which provides water for more than
1.4 billion people (Immerzeel et al., 2020).  The occurrence of natural hazards increases due to thawing permafrost (Zhang
and Wu, 2012, Yu et al., 2016), for example by destabilizing mountain slopes and rock walls (Deline et al., 2015). The
scientific and social importance leads to a stronger focus on permafrost areas, especially on the Tibetan Plateau (TP) where
permafrost conditions react fast to atmospheric warming (Cheng and Wu, 2007; Lu et al., 2017).
Permafrost research in engineering has a 60-year-long tradition on the TP (Chen et al., 2016; Yang et al., 2010). The
continuous use and life span of infrastructure depends on stable surface conditions which are strongly deteriorated by
permafrost degradation. The engineering corridors for infrastructure projects like the Qinghai-Tibetan Highway/Railway and
pipelines (Yang et al., 2010; Yu et al., 2016) were accompanied by monitoring permafrost sites based on borehole
temperature (Hu et al., 2020; Li et al., 2009a), ground temperature data (Cheng and Wu, 2007; Ma et al., 2006) and
geophysics: Small scaled ground ice distribution was investigated by ground penetration radar (Wang et al., 2020; Wu et al.,
2005; You et al., 2017) and by electrical resistivity tomography (ERT) (You et al., 2013, 2017) close to the important
Highways/Railways. Compared to the central and eastern parts of the TP permafrost surveys in the western and southern TP
are very scarce (Yang et al., 2010). Additional permafrost studies outside the engineering corridors are limited to modelling
results and large scale permafrost distribution maps (Ran et al., 2012, Cao et al., 2019, Obu et al., 2019). Implications of a
temperature warming followed by permafrost degradation for the entire TP are hard to deduce due to inadequate distribution
and small number of stations recording air temperature (Yang et al., 2010). Therefore, modelling approaches are gaining
increasing importance in order to estimate the consequences of the current temperature rise on the TP. This warming

temperature trend is reconstructed by δ 18 O records in four spatially well-distributed ice cores back to the beginning of the last century (Yao et al., 2006). Sun et al. (2020) confirm the relationship between the temperature increase and permafrost degradation on the TP by a slow adaption until the year 2100 based on a numerical heat conduction permafrost model. New statistical and machine learning approaches suggest that the permafrost extent on the entire TP is 45.9% (2003-2010) and they predict future permafrost degradation of 25.9% by the 2040s and 43.9% by the 2090s (Wang et al., 2019). Cheng and Wu (2007) also conclude that more than "half of the permafrost may become relict and/or even disappear by 2100".

This study aims to supplement the previously summarized studies with an assessment of probable occurrence of permafrost in remote high mountain regions unbiased by the location of the Tibetan corridorsand to provide a ground truthing for existing permafrost studies and maps on the TP. The use of the term "probable" is motivated by the fact that we do not have ground truthed temperature data for geophysical data validation. Furthermore, no small-scaled modelled permafrost distribution is available, and therefore we assess its occurrence indirectly. The spatial heterogeneity of our data (mapping, InSAR and ERT) and of topographic variations in permafrost occurrence also prevents us from providing precise elevational limits, thus we provide an assessment of probable occurrence of permafrost in a range according to the findings of the three methods.

Our study area (Figure 1, B and C) is located at the interface between continuous permafrost and seasonally frozen ground according to large-scale modelling results of PF-conditions on the TP (Sun et al., 2020). The location makes it a suitable environment to validate such large-scale models and to precisely define the interface with ground-truthed data. The validation is important, because the final conclusion would be that some higher region on the TP is not completely underlying permafrost conditions, unlike expected and modelled at other places at the TP (Cao et al., 2019; Ran et al., 2012)

The identification of periglacial landforms, subsurface ice and surface creeping rates on these landforms leads to an assessment of the probable occurrence of permafrost. The combination of field investigations and remote sensing techniques is a useful tool to detect permafrost occurrence (Bolch et al., 2019; Dusik et al., 2015; Monnier et al., 2014). Periglacial landforms such as active (creeping) rockglaciers and protalus ramparts can contain ice (Barsch, 1996; Scapozza, 2015; Schrott, 1996), and are considered indicators of permafrost occurrence (Frauenfelder et al., 1998; Haeberli et al., 2006; Kneisel and Kääb, 2007; López-Martínez et al., 2012). Especially on the TP only sparse literature is found that describes periglacial landforms in detail in combination with permafrost occurrence (Fort and van Vliet-Lanoe, 2007; Ran and Liu, 2018; Wang and French, 1995). However, these periglacial landforms as indicator for permafrost occurrence are essential for creating large-scale permafrost distribution maps (e.g. Schmid et al., 2015).

We present a multi-method approach to provide a reliable prediction of subsurface ice and permafrost occurrence to answer the following research questions:

- How are periglacial landforms distributed?

•    Do the investigated periglacial landforms like rockglaciers and protalus ramparts show an active status?

     •    Which creeping rates do the periglacial landforms indicate?

We created (A) an inventory of periglacial landforms indicating potential subsurface ice-occurrence, we (B) acquired Electrical Resistivity Tomography (ERT) data to validate the ice occurrence of selected landforms, and we (C) then used multi-annual surface creeping rates from InSAR time series analysis to corroborate the hypothesis of long-term ice

occurrence due to permafrost conditions above a special elevation. As a result, the study provides probable occurrence of permafrost by combining these three methods for a catchment in an high-altitude mountain range of the TP

## 2 Study area

The Western Nyainqêntanglha Range (Figure 1) was formed during the Himalayan-Tibetan orogenesis as part of the central Lhasa block (Kapp et al., 2005; Keil et al., 2010). From Tertiary to Quaternary, the Nyainqêntanglha area was controlled and

compressed by a fracture belt which folded and rose violently, forming the Nyainqêntanglha Mountains, with the highest peak of 7162 m a.s.l. (Kidd et al., 1988; Keil et al., 2010). Our study area, the Qugaqie catchment is characterized by Cretaceous red beds and sandstone in the northern part and by early tertiary granodiorites in the center. The bedrock of the southern part consists of biotite adamellites and glaciers in the highest zone (Kapp et al., 2005; Yu et al., 2019). The atmospheric circulation pattern and the topographic characteristics are responsible for a similar glacier distribution pattern in

all North-oriented catchments of the Western Nyainqêntanglha range, including the Qugaqie Basin (Kang et al., 2009; Bolch et al., 2010). On the Lee side of the main Western Nyainqêntanglha crest and therefore in the Lee site of the moisture of the Indian Summer Monsoon (ISM) the glaciers are smaller in area and length (Bolch et al., 2010) (Figure 1, B). Bolch et al. (2010) also investigated the glacier shrinkage based on satellite data. They observed a glacier retreat of about −9.9±3.1% between 1976 and 2009. Zhang and Zhang (2017) observe a melting rate −0.30 ± 0.07 m yr$^{-1}$ over the entire western

Nyainqêntaglha range from 2000 to 2014. The Zhadang glacier located in the Qugaqie head lost an area of almost 0.4 km² in the same time span and covered an area of 2.36 km² in 2009. The corresponding retreat rate is 14 %, slightly larger than the regional average, which could indicate a slightly faster deglaciation of the smaller, north-orientated glaciers in the Western Nyainqêntanglha range.

The Qugaquie catchment is a sub-catchment of the Nam Co catchment, which is influenced by a strong climate seasonality

driven by different wind systems throughout the year (Yao et al., 2013): Westerlies dominate in the winter months and provide cold, dry continental air from east to northeast (Figure 1, A, blue arrows), with temperature minima below -20 °C. The dry season ends with the onset of the ISM (Figure 1, A, red arrows), which provides moisture from May to September (Mügler et al., 2010). 80% of the annual precipitation (295–550 mm/y) occurs during the monsoon dominated summer months (Wei et al., 2012). The influence of the East Asian Monsoon on our study area is minor but it is an important source

of moisture for the eastern TP (Figure 1, A, black arrows). Consequently, the study area of the Qugaqie Basin, situated in the

Western Nyainqêntaglha Range (Figure 1, B), is characterized by semiarid climate and a large amount of solar radiation due to the high elevation and reduced cloud cover (Li et al., 2009). With an area of almost 60 km² the basin drains into the dimictic lake Nam Co (Figure 1, B) and the relief extends from 4722 m a.s.l. to an elevation up to 6119 m a.s.l.

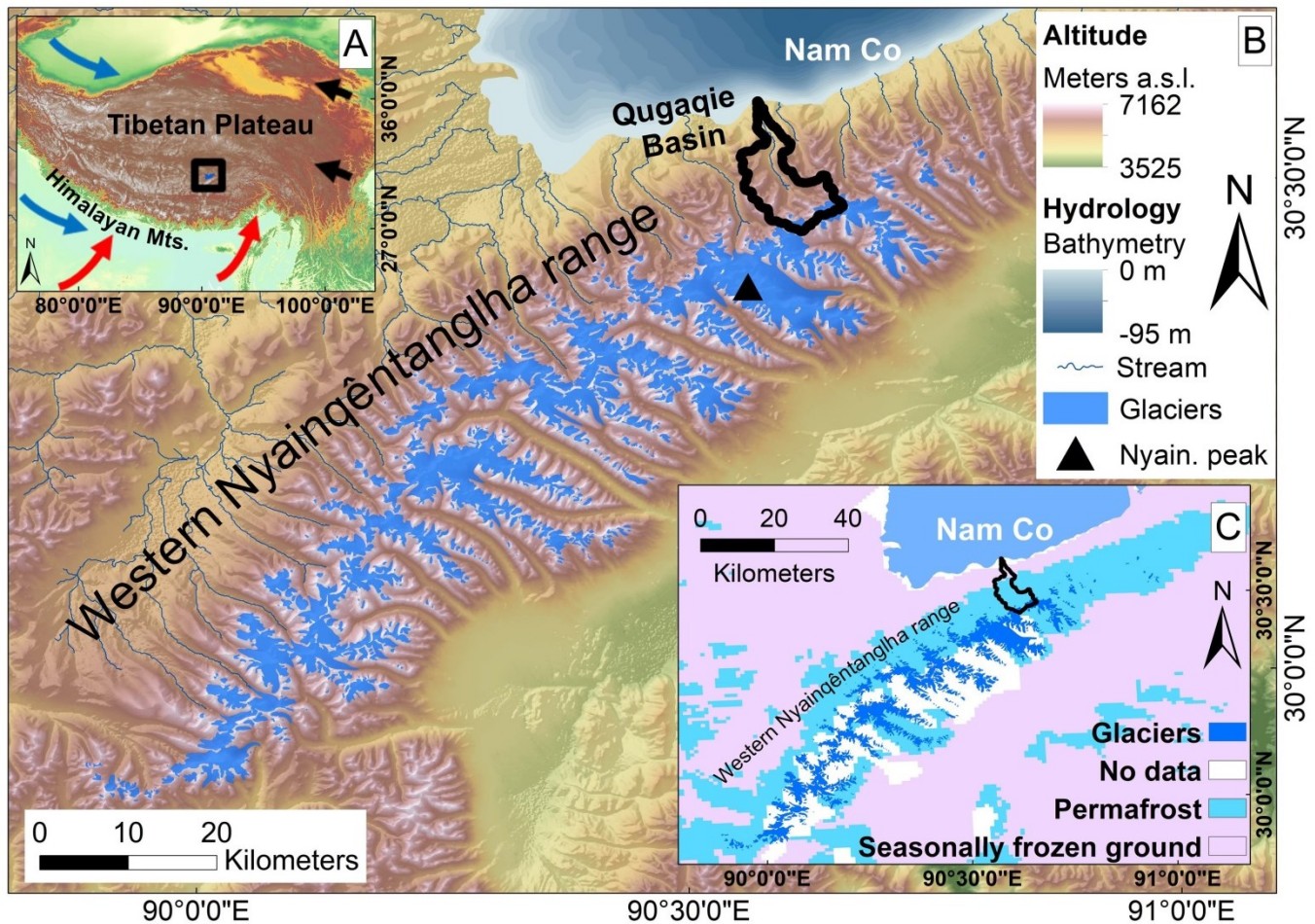

 **Figure 1: A: Location of the study area within the Tibetan plateau (TP) (Background on SRTM DEM v4; Jarvis et al., 2008). Different wind systems influencing climate of the TP are shown by blue (Westerlies), red (Indian Summer Monsoon) and black (East Asian Monsoon) arrows based on Yao et al. (2012). B: Overview map of the Nam Co catchment with the altitude colors and the study area of the Qugaqie catchment (thick black lines). Note the greater glacier extents on the south-oriented mountain range. Bathymetric data originated from (Wang et al., 2009) (Hillshade and DEM background based on SRTM DEM v4; Jarvis et al., 2008). Glacier extents originated from the GLIMS database (Cogley et al., 2015; Guo et al., 2015; Liu and Guo, 2014). C: Permafrost distribution in the Western Nyainqêntanglha range based on Zou et al. (2017).**

Detailed information about permafrost occurrence and distribution in the study area is very scarce. Tian et al. (2006) determined a lower limit of permafrost based on soil probes at an elevation of around 5400 m a.s.l. along the northern slopes of Mt. Nyainqêntanglha peak (Figure 1, B). This is generally higher than in other regions (>4500 m a.s.l.) of the TP (Ran et al., 2012). Schütt et al. (2010) sampled lacustrine sediments from a permafrost lens in an outcrop at the Gangyasang Qu's

entry into the North-western end of Lake Nam Co at 4722 m a.s.l. Zou et al. (2017) distinguish between seasonally frozen ground and permafrost on their distribution map over the TP (Figure 1, C). According to their map permafrost is existent at elevation higher than 5000 m a.s.l. and covers more 90% of the than study area. The visible data gaps were not further discussed by Zou et al. (2017). A coarse overview including a distinction between glacial and periglacial processual states

around the lake Nam Co is given by Keil et al. (2010). A two-year temperature-dataset on the Zhadang glacier, recorded at 5680 m a.s.l. by an automatic weather station (2009-2011) in 2 m height, shows a mean annual air temperature (MAAT) of - 6.8°C (Zhang et al., 2013) and suggests permafrost conditions for the surrounding periglacial landscape.

## 3 Data and Methods

We have used three different methods (A-C) to gain insights into permafrost-indicating periglacial landforms and to assess
the lower occurrence of probable permafrost in the Qugaqie catchment. The following methods (Figure 2) indicate information about permafrost conditions:

(A) Geomorphological Mapping: A map visualizes the distribution and characteristics of periglacial landforms and geomorphometric features.
(B) Geophysical methods: Electrical resistivity tomography (ERT) identifies ice content and reveals the subsurface structure
of periglacial landforms.
(C) Microwave remote sensing: Interferometric Synthetic Aperture Radar (InSAR) time-series analysis of ESA's Sentinel-1 satellite data detects perennial, constant creeping rates of active periglacial landforms.

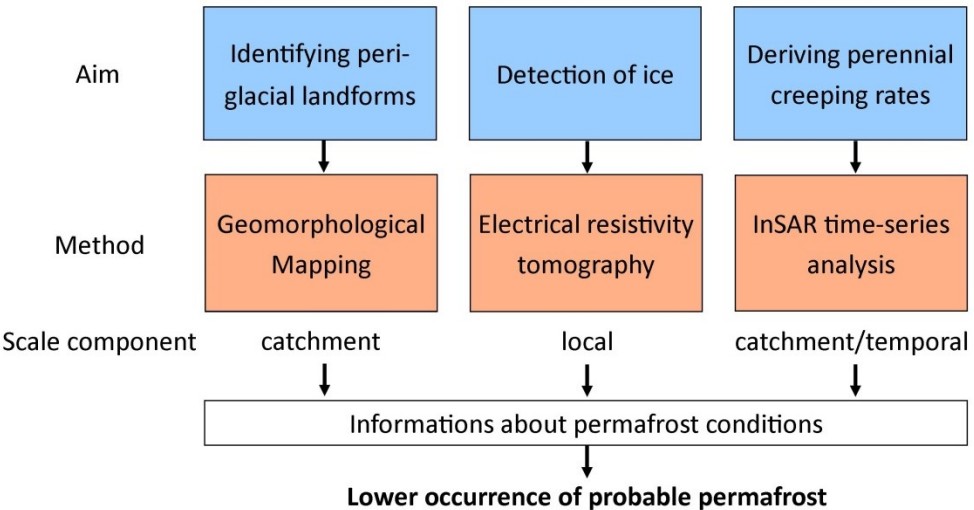

**Figure 2: Schematic workflow of applied methods to assess lower occurrence of probable permafrost.**

(A) A geomorphological map visualizes the distribution and characteristics of landforms and geomorphometric features with the focus on periglacial landforms on a catchment-wide/regional scale. Periglacial landforms like rockglaciers (Barsch,

1996) and protalus ramparts (Scapozza, 2015) can potentially preserve ice over a long period of time (Ballantyne, 2018) and their activity and perennial creeping is an indicator for permafrost occurrence (Delaloye et al., 2010; Eckerstorfer et al.,

2018; Esper Angillieri, 2017). This circumstance is validated (B) by ERT to detect subsurface ice on a local scale. (C) InSAR time series analysis detects perennial creeping which is typical of active periglacial landforms. The permafrost occurrence is indicated by activity of landforms and the corresponding surface structures like bulges, furrows, ridges or lobes We make use of the fact that the deformation of debris supersaturated with ice causes surface displacement by downwards permafrost creep (Barsch, 1996; Delaloye et al., 2010). Therefore, we concretize surface displacement (rates) as permafrost

creep (creeping rates) in this study. Although the continuous movement of periglacial landforms and the presence of ice can be implied from InSAR data alone, ground truth at selected locations by ERT is essential to exclude other possible interpretations.

We assess the lower occurrence of probable permafrost by the mean altitudinal distribution of periglacial landforms, by the subsurface ice occurrence which has been validated with geophysics, and by the active status which is indicated by perennial

surface creeping rates (Figure 3). An occurrence of sporadic Permafrost is not excluded in lower elevation, but cannot be validated by the used methods and due to scale issues.

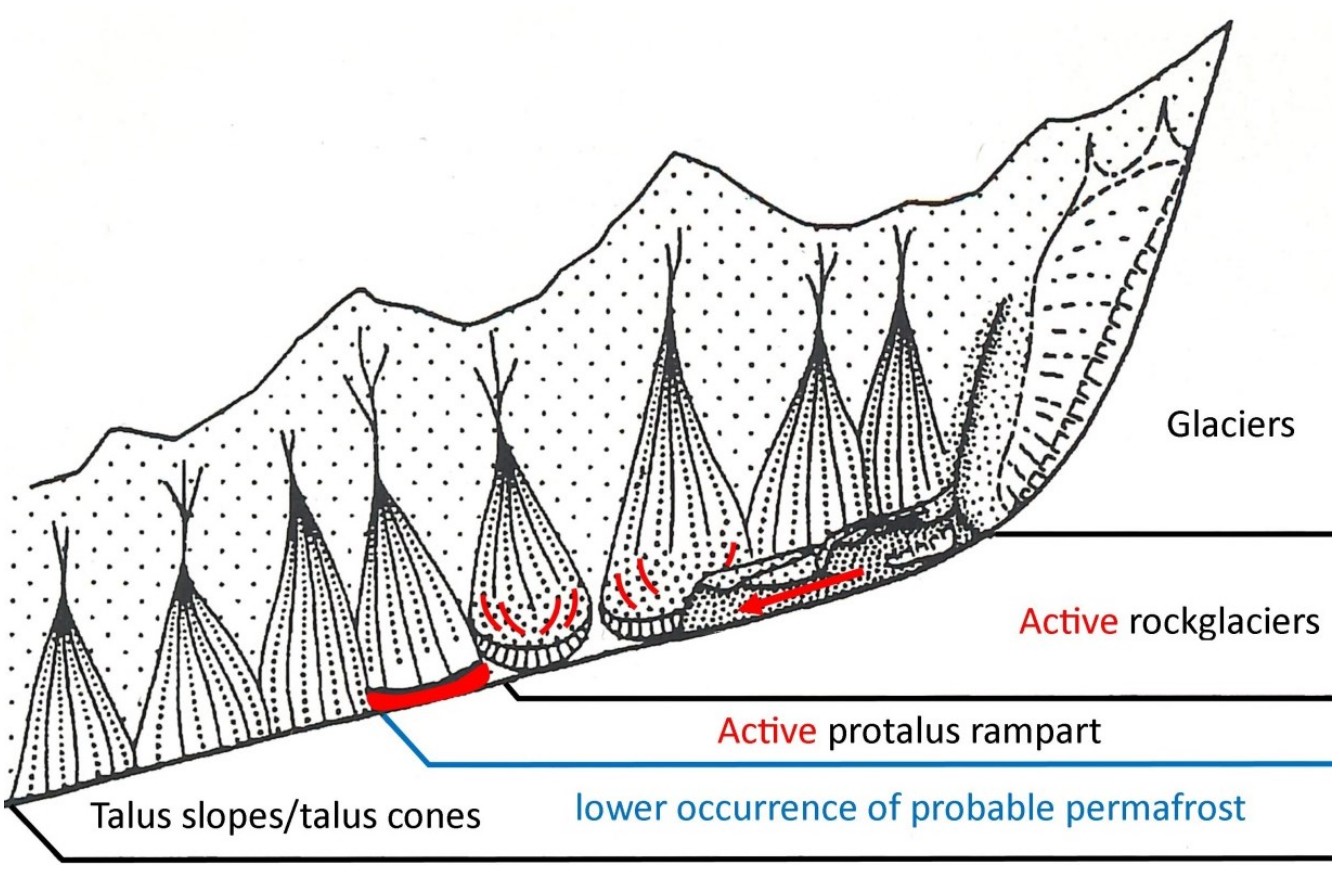

**Figure 3: Schematic, hypsometric distribution of mapped landforms. Red features show active, multiannual creeping structures (furrows, lobes, bulges, ridges) of periglacial landforms indicating the lower occurrence of probable permafrost. Modified from Barsch (1996) after Höllermann (1983).**

### 3.1 Inventory of cryospheric mesoscale landforms

The mapping procedure consists of the elementary mapping steps, described by Knight et al. (2011) and Otto and Smith (2013). Pre-Mapping includes analyses of digital elevation models (DEM) and mapping of landforms on optical images in a scale of 1:10 000 (named here as Mesoscale following to Höllermann (1983)). The DEM used in this study originates from TanDEM-X data (2015) with a resolution of 12 m (©DLR). The optical images are based on Digital globe, BING maps (2013) and Google Earth data (2007-2012). Geomorphological symbols were used after Kneisel et al. (1998) for field mapping and after Otto (2008) for the digitized visualization in ArcGIS. During the field campaign, the main focus was on the mapping of periglacial landforms in the Mesoscale (Höllermann, 1983). These landforms are components of the periglacial zone which is defined by seasonally-frozen and perennially-frozen ground (French, 2017). A differentiation between seasonally-frozen and perennially-frozen movement behavior is given by the InSAR data and a derived model by Reinosch et al. (2020). This data was used for the preparation of the cryospheric landform identification. Next to optical and InSAR data, the periglacial landforms were identified in the field by an inspection of the form, the substrate, the catchment and the potential process which formed the landform. The results section describes the inventory statistically and includes morphological field observations which could not be included in the map due to scale issues. For example, small-scaled death ice holes were not included in the mesoscale geomorphological map. During post-mapping we integrated the field-mapped information into ArcGIS. Additional features like a stream network, lakes, ridges, glacier extents and moraines were delineated with the help of the mentioned DEM, a hillshade map (azimuth 315°, altitude 45°) and the mentioned optical images. Glacier extents were digitized based on optical images of the year 2013 (BING maps). Rockglaciers were identified following the comprehensive description by Barsch (1996): If the form shows a tongue- or a lobate shape in the field and at the optical images, we classified the landform as a rockglacier. Additionally, field observations like coarse clasts at the surface and at the front indicate typical rockglacier substrate. Protalus rampart are classified by a coarse debris accumulation in front of a rock wall. A small depression occurs between the non-lobate buldge and the weathering rockwall. We followed the geomorphological mapping approach based on the baseline concepts (V 4.0) of the IPA Action Group "Rock glacier inventories and kinematics" (Delaloye et al., 2018; Delaloye and Echelard, 2020) and mapped the extended geomorphological footprint of the rockglaciers. Additional mapping criteria of rockglaciers in the field were visible creeping structures on the surface (ridges, furrows, and lobes as those shown in Figure 4, A).

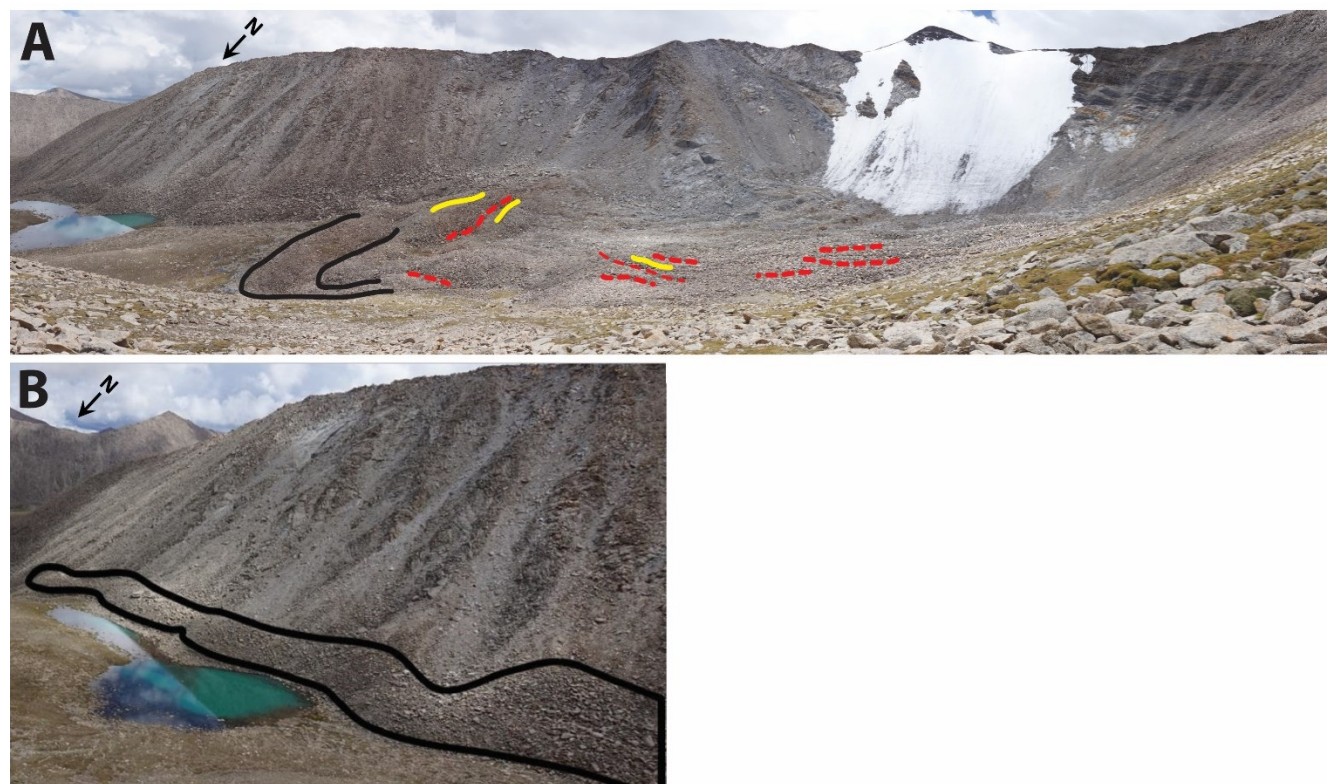

**Figure 4: (A) Panorama view on a rockglacier (No. 1) with marked creeping structures (lobe in black, ridges in yellow, and furrows in dashed red) in the Qugaqie Basin in hanging valley 3. (B) Example of a protalus rampart in hanging valley 3 of the Qugaqie Basin. A bulge (in black) formed through creeping of rockfall deposits. The length of the bulge is approximately 500 m. The location of the photos can be found in Figure 6 B. (photos: J. Buckel)**

Protalus ramparts (Figure 4, B) were mapped as periglacial features or permafrost-related landforms as suggested by Scapozza (2015). A straight headwall for the sediment source is required, as the sediment originated by rockfalls and is accumulated at the foot of the rockwall. Infiltrating moisture originating from precipitation and snow melt freezes the sediment deposit and creates a bulge parallel to the rockwall. These ice-permeated rockfall deposits creep downwards. Scapozza (2015) also noted the challenge to differentiate protalus ramparts from initial talus rockglaciers in the sense of Barsch (1996). Protalus ramparts mapped in the present study show no ridges, furrows, or lobes at the surface, but the mapped rockglaciers do. It is pertinent to point out that our mapping procedure both in the field and during post-mapping consistently differentiates between rockglaciers and protalus ramparts based on the above-mentioned criteria. An incorrect determination as pronival ramparts can be minimised by the absence of longer existing snow fields due to arid climate conditions during the winter and the strong solar radiation and less cloud cover due to the extreme altitude (compare Hedding, 2016).

## 3.2 Ice detection by ERT

Electrical resistivity tomography (ERT) is a widely-used method in geomorphology (Schrott and Sass, 2008). The application works especially well for subsurface ice detection due to strong differences between frozen (high resistivity values) and unfrozen ground (low resistivity values) (Hauck and Vonder Mühll, 2003; Hauck and Kneisel, 2008). Since the end of the 1990ies the method has been established for permafrost detection in solid rock (Krautblatter et al., 2010; Hartmeyer et al., 2012) and in debris-ice mixtures, like rockglaciers (Von der Mühll et al., 2002; Kneisel et al., 2008; Rosset et al., 2013; Emmert and Kneisel, 2017, Mewes et al., 2017).

For the usual four-point measurement of the ground electrical resistivity, two electrodes feed current into the ground, which establishes an electric field in the subsurface. Another pair of electrodes is used to measure the voltage drop between two other locations on the surface. In order to obtain information on the two-dimensional distribution of electrical resistivity in the subsurface, a linear arrangement of the four electrodes is used to measure at different positions along the profile and with varying distances between the electrodes (Wenner array). The apparent resistivity (in $\Omega$m) of each measurement can be calculated from the injected current, the applied voltage, and a factor, which takes the geometry of the arrangement into account. Subsequently, inverse modelling techniques are used to reconstruct the resistivity structure of the subsurface from the measured apparent resistivity data (Loke and Barker, 1995).

We performed ERT measurements during a field campaign in July 2018. We worked with multi-electrode (50) equipment "GeoTom-MK" (GEOLOG2000, Augsburg, Germany), a maximum spacing of 2 meters, allowing a maximum profile length of 98 m with a single measurement. To obtain longer sections, we used the roll-along procedure illustrated in Figure 5. For this procedure, two cables were available (denoted A and B), each equipped with 25 channels. First, both are connected with the control unit to obtain pseudosection number 1 (Figure 5). Next, cable B (and all connected electrodes) remains at the same location, whereas cable A is moved to the right of cable B to measure the Preudosection number 2, and so on. The location of the ERT profiles was partly constrained by logistical conditions. Due to the high altitude, the crew had to stay at one level for three days to get adapted to altitude. The measurement locations were not accessible by vehicles, and a few hours were needed every day to reach the sites, resulting in limited productivity. Therefore, we tried to locate the profiles efficiently to obtain a representative data set of the valley. We covered different landform features (moraine, valley bottom, Rockglacier) where permafrost conditions were assumed. Blocky surfaces constitute a challenge for ERT measurements due to instability and a lack of fine material necessary to provide sufficient contact for the electrodes. In cases where no soil material could be found that closed the gaps between the boulders, we inserted the end of each electrode into a sponge saturated with salt water to improve connectivity to the fine material. The saturated sponge kept the fine material wet and diminished desiccation through high solar radiation. The ERT-data was processed with the Res2Dinv-Software (©Geotomo Software).

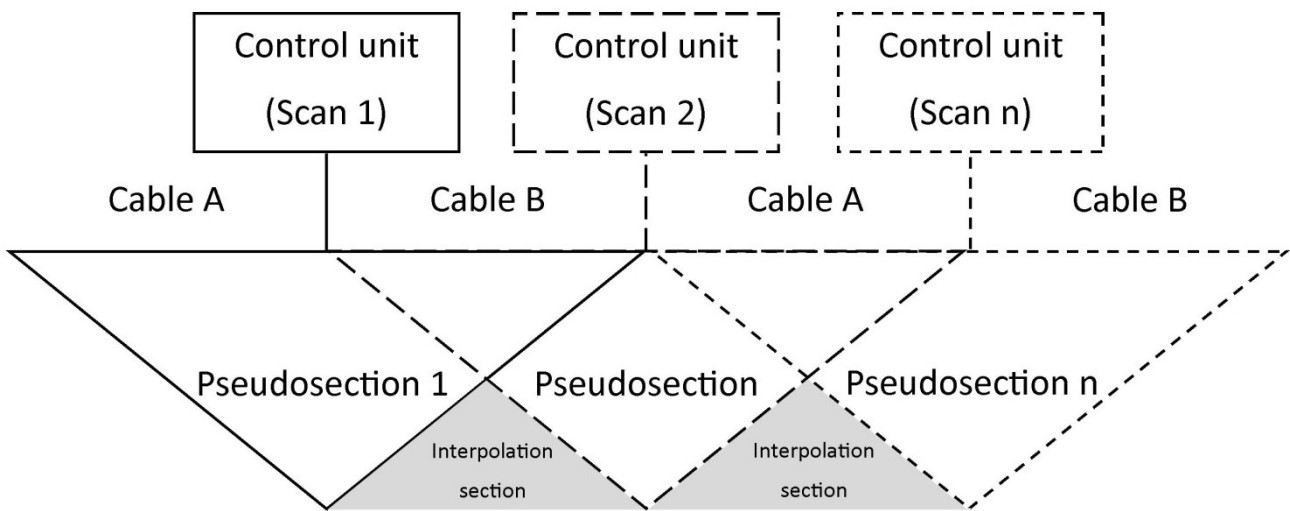

Figure 5: Measurement setup for the Roll-along procedure (adapted from N El Sayed et al., 2018).

### 3.3 Creeping rates by InSAR-analyses

InSAR time series analysis is an active microwave remote sensing technique, which can exploit the phase change of the backscattered microwaves to determine relative surface displacement in the order of millimetres to centimetres (Osmanoğlu et al., 2016). Both the amplitude and the phase of the microwave backscatters are used for InSAR. After precisely co-registering all acquisitions, it is possible to calculate the average phase change of each resolution cell over time, which contains a number of different signals, including whether a resolution cell moved closer to the receiver, i.e. the satellite, or further away from it. These images of phase change are called interferograms. The accuracy of the derived motion is dependent on a number of different factors, including the frequency of the emitted wave, the atmospheric delay, the accuracy of its modelling, the topographic data used to correct the images, the choice of reference points, the surface characteristics of the observed structure and the frequency of the data acquisitions (Hu et al., 2014).

The reliability of an interferogram is often described by its so-called coherence. Coherence is a measure of phase stability with a value near zero representing poor reliability and values near one representing high reliability (Crosetto et al., 2016). If the backscatter characteristics of the observed surface change too much between two acquisitions, e.g. due to snow cover, vegetation or events occurring between the acquisitions like rock falls, the coherence is poor and no phase change can be determined reliably. Coherence also decreases with increasing displacement and displacements larger than half the SAR wavelength (~2.8 cm for Sentinel-1) cannot be determined accurately. For this study we chose a coherence threshold of 0.3 and discarded areas with coherence values below 0.3. This threshold is similar to the one chosen by Sowter et al. (2013) and provides good spatial data coverage while also excluding unreliable data. The issue of low coherence or decorrelation is exacerbated for interferograms with a long temporal baseline i.e. a long time period between data acquisitions. No Sentinel-1

data is available for a period of 48 to 96 days during the summers of 2016 and 2017. These longer temporal baselines cause decorrelation during the summer months on some of the faster landforms. Freezing and thawing of the ground leads to reduced coherence values in autumn and spring. The coherence over periglacial landforms in the Qugaqie Basin is relatively good, due to the lack of high vegetation on actively moving landforms and the relatively sparse snow cover in winter visible on optical Sentinel-2 acquisitions.

Exploiting the phase change with InSAR provides only relative surface motion towards the satellite or away from it. The Line-Of-Sight (LOS) of the satellite is therefore very important, as motion with a very different direction compared to this LOS is severely underestimated (Hu et al., 2014). The severity of this underestimation depends on the angle between the LOS and the direction of the surface displacement. An angle close to 0° will cause only minor underestimation, while displacement with a direction near 90° to the LOS will be severely underestimated or even completely overlooked. The

Sentinel-1 satellites follow a circumpolar orbit and observe the earth obliquely with an incidence angle of 33°-43° (Yague-Martinez et al., 2016). Both ascending (satellite travelling south to north) and descending (satellite travelling north to south) acquisitions are therefore sensitive to vertical surface displacement and towards the East or West respectively, but very insensitive to displacement towards the North or South. We always select the geometry with the highest sensitivity towards the expected displacement direction to calculate our displacement and velocity results.

The surface displacement data presented in this study represents a spatial subset of a surface displacement model originally based on Reinosch et al. (2020). For our analysis of the Qugaqie Basin, we processed 278 interferograms from 74 ascending acquisitions (June 2015 to December 2018) and 257 interferograms from 63 descending acquisitions (November 2015 to December 2018) (Table 1). The temporal baselines, i.e. the time period between two data acquisitions, of individual interferograms is mostly 12 to 36 days with a maximum of 72 and 96 days for ascending and descending orbits, respectively.

All data acquisitions originate from ESA's Sentinel-1 a/b satellite constellation. Both ascending and descending datasets were processed using Small Baseline Subset (SBAS) time series analysis (Berardino et al., 2002), with a coherence threshold of 0.3. Mean velocities were calculated by dividing the cumulative displacement observed during the observation period by the length of the observation period (2015-2018).

**Table 1: Summary of ISBAS processing parameters**

| Geometry | Observation period | Acquisitions | Interferograms | Temporal baseline | Coherence threshold |
|---|---|---|---|---|---|
| ascending | 2015-06-05 to 2018-12-22 | 74 | 278 | 12 to 72 days | 0.3 |
| descending | 2015-11-15 to 2018-12-29 | 63 | 257 | 12 to 96 days | 0.3 |

All surface velocity data of periglacial landforms has been projected along the direction of the steepest slope under the assumption that the motion of the described landforms is mainly gravity-driven by an ice-debris mixture. Hereafter we will refer to the mean surface velocity of periglacial landforms projected along the steepest slope as "creeping rates" to reflect this assumption. We calculate a sensitivity coefficient to compensate for the underestimation of the displacement signal caused by the disparity between the LOS and the assumed displacement direction. We followed an approach developed for the study of landslides (Notti et al., 2014), as the displacement of landslides is gravity-driven, which we also assume to be true for the periglacial landforms investigated in this study. Creeping rates presented in this study were not verified by independent measurements (GPS measurements, laser scans, optical remote sensing etc.), as no such data sets exist for our study area. Reference points are located on bedrock whenever possible and on ridges or stable, vegetated moraines with good coherence if no coherent bedrock was available (compare Figure 9, A). Areas which are likely unmoving on a multiannual scale, such as the old moraines at the entrance of the Qugaqie Basin, display LOS velocities of ±2.4 mm/yr during our observation period. This does not provide information regarding the accuracy of the seasonal variations of our surface displacement results but it indicates that the multiannual LOS velocity results are reliable. We use this variation of ±2.4 mm/yr over likely stable areas as the precision of the mean LOS velocity during our observation period. The precision of the creeping rates was determined by dividing the precision of the LOS velocity by the sensitivity coefficient. It therefore varies between 2.4 mm/yr and 12.0 mm/yr for areas with a sensitivity coefficient of 1 and 0.2 respectively (Reinosch et al., 2020).

## 4 Results and Interpretation

### 4.1 The cryosphere of the Qugaqie Basin

The geomorphological map in Figure 6 shows features of the mesoscale cryosphere in the Qugaqie Basin: glaciers, moraines, protalus ramparts and rockglaciers. The moraine distribution suggests that former glaciers extended to the present shoreline of the Nam Co at their largest size during Marin Isotope Stage (MIS) 3 (Dong et al., 2014). Multiple smaller moraines are displayed in closer proximity to today's glaciers (Figure 6). Glacial landforms like valley glaciers, cirque and wall glaciers increase in number and size towards the south due to a higher elevation and shorter distance to the main ridge (Figure 6). Only the Genpu (1.56 km²) and the Zhadang (1.41 km²) glaciers are considered valley glaciers, most of the other glaciers are located in the head of the hanging valleys as cirque glaciers. The northward orientation of all glaciers is a result of the Lee-effect towards incoming moisture from the southern direction. The topographic barrier of the Western Nyainqêntanglha Range detains precipitation and causes an asymmetric and uneven north-south distribution of glacier extents expressed by smaller extents in the northern catchments draining in the Nam Co like Qugaqie (compare Bolch et al., 2010). The glacial zone with a cumulative glacier area of 4.07 km² (bing maps, 2013) extends from 5500 m a.s.l. to the highest elevation (6086 m a.s.l.) with a mean elevation of 5770 m a.s.l.

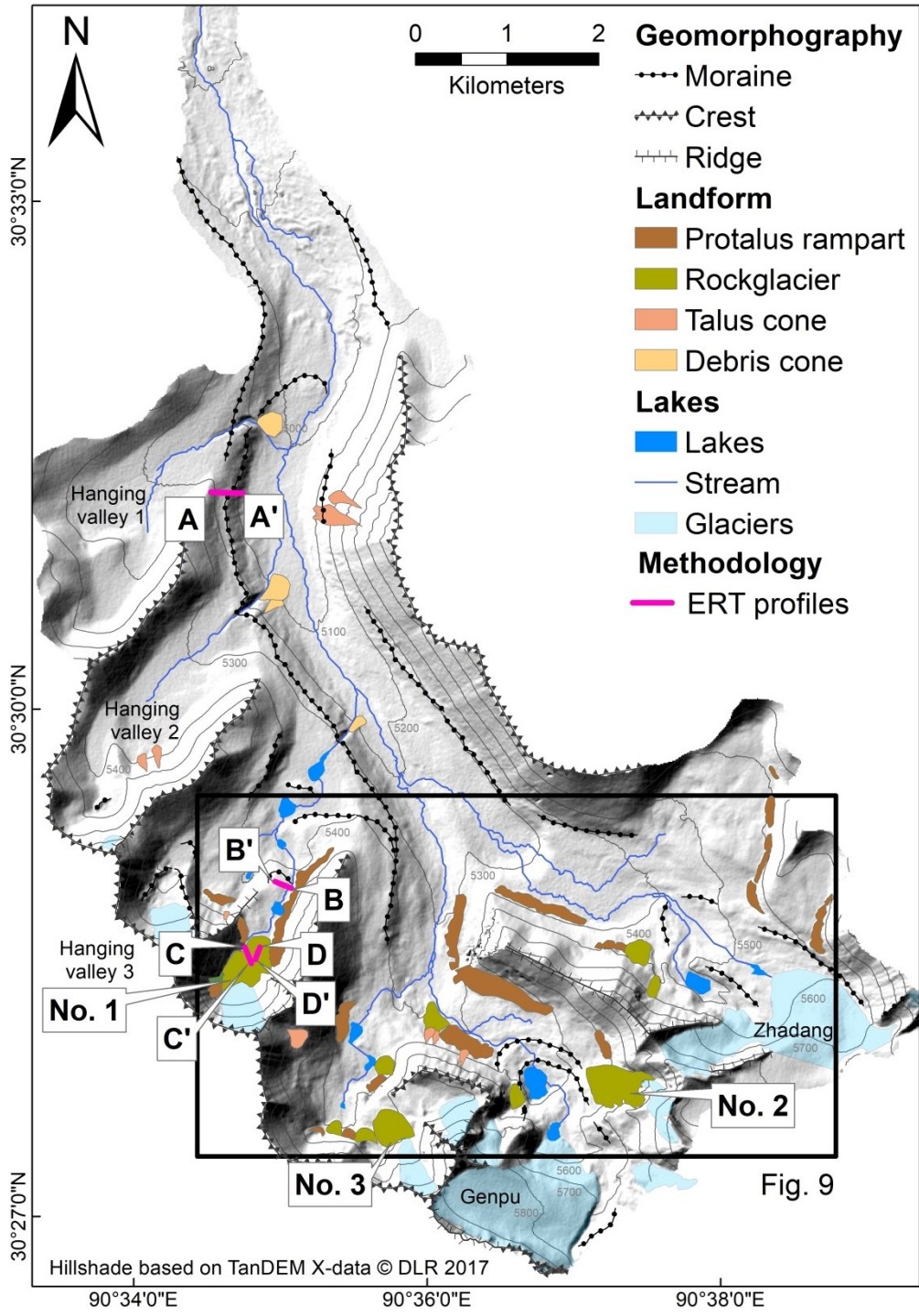

**Figure 6: Geomorphological map of the Qugaqie basin. The locations of the ERT profiles are shown with purple lines. Periglacial landforms are greenish (rockglaciers and protalus ramparts). The black rectangle represents the boundary of the map shown in Figure 9.**

The altitudinal (mean) landform distribution illustrates the statistical analyses and displays a typical high-mountain pattern (Figure 7). Debris and talus cones can be found in lower altitudes. The periglacial landforms (i.e. protalus ramparts and rockglaciers) are located between elevations of 5300 m and 5600 m a.s.l. and the average number of periglacial landforms is situated around 5500 m a.s.l. We conclude from this altitudinal distribution a probable occurrence of permafrost higher than 5300 m. a.s.l., which has to be supported by validating ice occurrence and the status of activity of these landforms.

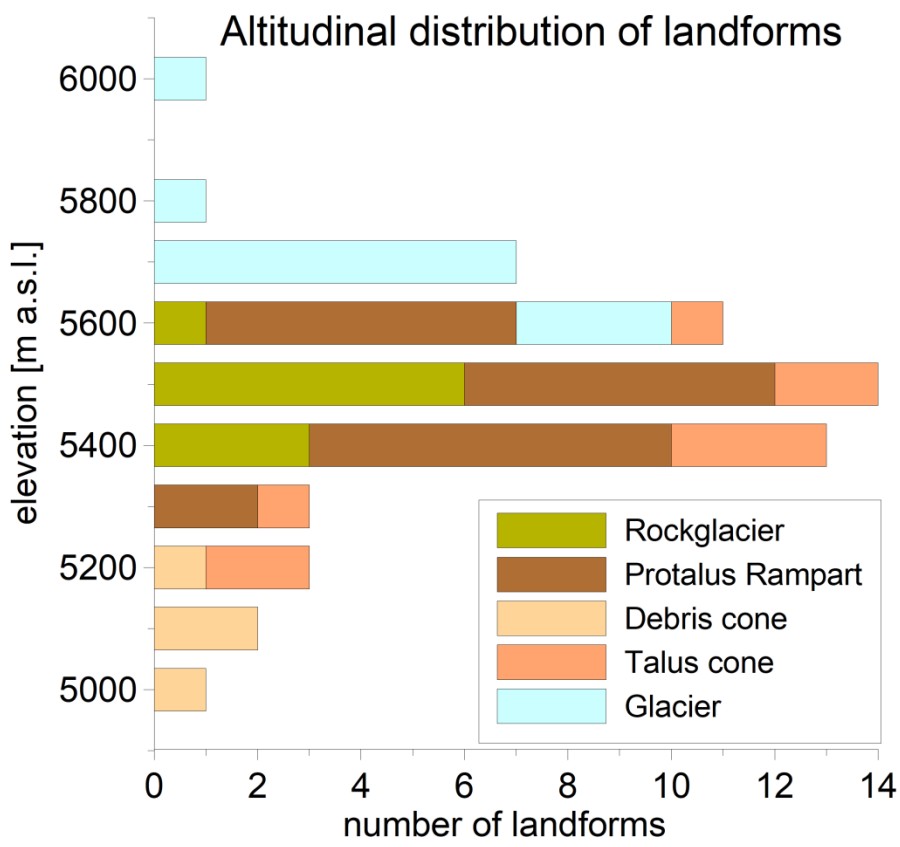


**Figure 7: Altitudinal (mean) landform distribution of the Qugaqie basin derived from the landform inventory.**

     Most rockglaciers are located in cirques and three are supplied by glacial melt water resulting in greater extents compared to rockglaciers without a glacier in their catchment (Figure 6, No. 1, 2 and 3). Additionally, moraine deposits, talus slopes and protalus ramparts provide the sediment accumulation at the base required for the formation of a rockglacier besides water

availability (Knight et al., 2019). The altitudinal distribution of the rockglaciers extends from 5363 m to 5789 m a.s.l. with a mean elevation around 5500 m a.s.l. (Figure 7, Table 2). Rockglacier surfaces display clear creep structures and rockglacier-typical bulges, furrows and lobes (Figure 4, A). There is no pronounced lichen growth, and the uppermost material is extremely unstable. These field observations in combination with the observed creeping rates (Figure 9, B) allow the conclusion of an active status of the rockglaciers, which indicates ice occurrence and, thus, permafrost conditions (according

to Barsch, 1996). The altitudinal distribution of protalus ramparts has a narrower range of min-max values, but they are located at a similar mean elevation. The mean area of the individual protalus ramparts is only half of the mean area of the individual rockglaciers, i.e., protalus ramparts are generally smaller than rockglaciers (Table 2, Figure 6), but there are twice as many. Protalus ramparts are situated in front of rocky slopes and are characterized in contrast to rockglaciers by a shorter dimension in down slope (Figure 4 and Figure 6).

The mesoscale periglacial landforms (mean elevation) are situated between 5300 and 5600 m a.s.l. This altitudinal distribution serves as one component of the three methods for assessing the probable occurrence of permafrost in the catchment.

Table 2: Statistical description of cryotic landforms based on DEM-analyses.

| | No. | cumulative area [m²] | area (mean) | elevation (min, max) | elevation (mean) |
|---|---|---|---|---|---|
| Protalus rampart | 22 | 1018014 | 46273 | 5292, 5685 | 5530 |
| Rockglacier | 10 | 830185 | 83019 | 5363, 5789 | 5523 |
| Glacier | 11 | 4075580 | 370507 | 5504, 6086 | 5771 |

**4.2 ERT-based ice detection**

ERT is a common method to detect ground-ice in the subsurface, inferring permafrost conditions (Lewkowicz et al., 2011), if ground ice is present for two consecutive years. With the help of ERT we were able to provide evidence for the existence of ground ice at specific test sites. Figure 6 displays the locations and indicates an altitudinal increase of the four ERT-profiles (A to D). The measured resistivity values were compared with tables by Hauck and Kneisel (2008) and Mewes et al. (2017).

These studies also address ice detection in high altitude periglacial environments. Table 3 sums up our measured resistivity values and classifies the values in terms of material characteristics. Different studies show resistivity values of till in a range from 1 to 10 kΩm (Reynolds, 2011), from 5 to 10 kΩm (Thompson et al., 2017) and from 50 – 100 kΩm (Vanhala et al., 2009). The diversity of resistivity ranges and the resulting non-uniqueness can be overcome by using additional methods to support the final conclusions.

**Table 3: Resistivity values for different materials derived by field measurement. The used terms of the interpreted material followed Hauck and Kneisel (2008) and Mewes et al. (2017)**

| Material | Resistivity [kΩm] |
| --- | --- |
| Sandstone (moist – dry) | 0.5 – 5 |
| Till | 20– 80 |
| Unfrozen sediment (moist – dry) | 1 – 20 |
| Ice-poor permafrost (ice lenses, ice-interspersed till ) | 50 – 150 |
| Ice-rich permafrost (massive ice body) | 150 – 4 000 |

Profile A (Figure 8) ranges from 5090 to 5230m and represents subsurface conditions in the lower altitudinal areas of the catchment, for example in a lateral moraine. At the surface the profile has a length of 348 m, but the length information in the following text refers to the x-axis which corresponds to planar 2D-view (the topographic effect is not displayed). From ~120 m on, we observe a slope-parallel highly resistive layer (highlighted by the black line in Figure 8, A) with resistivity values ranging between 5 and 100 kΩm and an average thickness of 10 m. We interpret this layer as compressed till without ice content, based on the resistivity range, the compressed glacial sediment accumulation and the absence of creeping structures indicating ice. According to Yu et al. (2019) the underlying bedrock consists of sandstone, which explains the low resistivity values below the resistive moraine deposits. Between 0 and 20 m along the profile, the electrodes were directly attached to the outcropping, weathered sandstone. The resistivity values around 5 kΩm correspond to dry sandstone bedrock, which is exposed to strong solar radiation. The hydraulically impermeable till cover is not present between 20 and 120 m, and moisture infiltrates as slope water saturating the sandstone bedrock underneath the moraine and decreasing electrical resistivity.

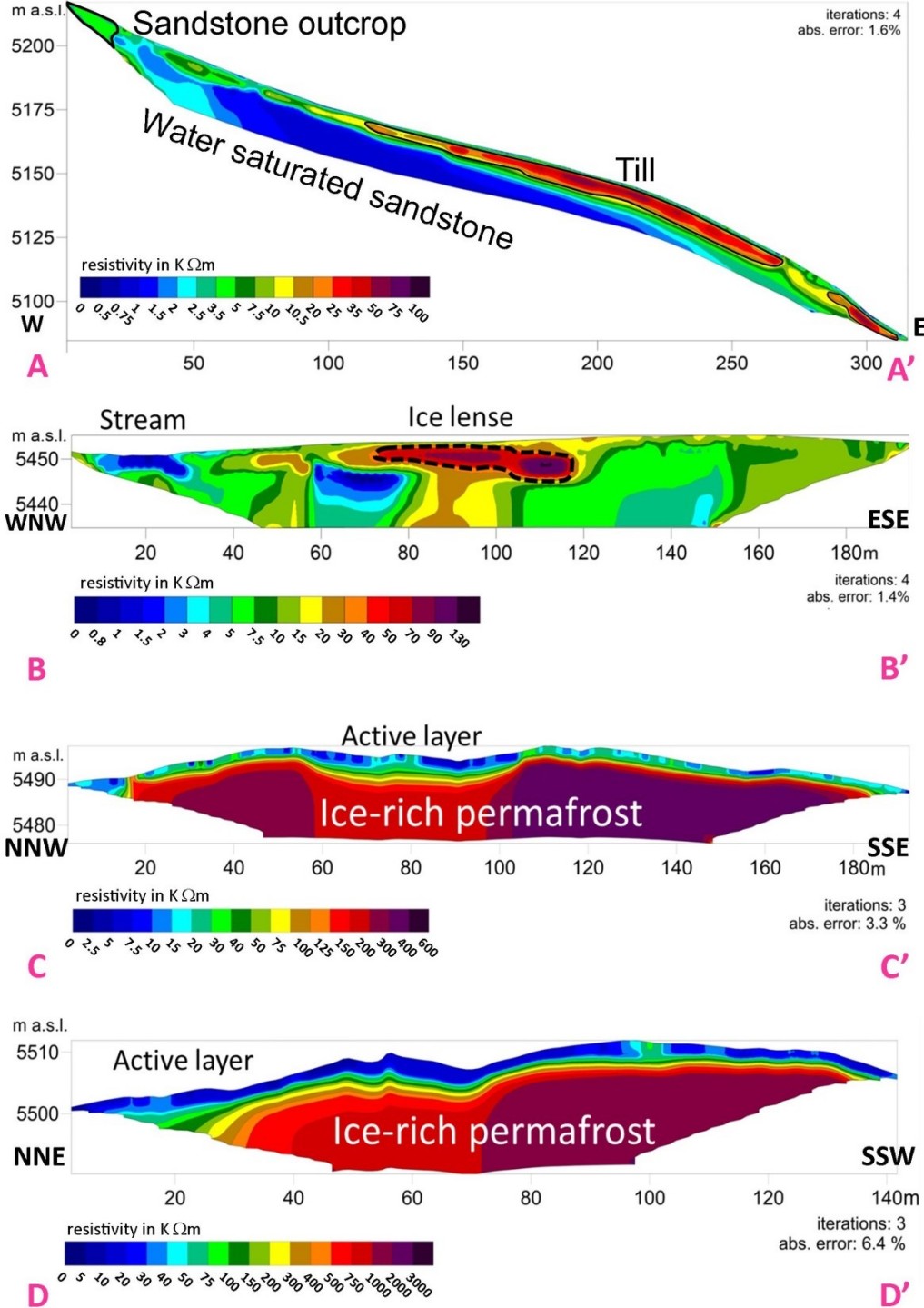


**Figure 8: Electrical resistivity sections along the four ERT profiles recorded in July 2018 with a standard spacing of 2 m. Profile C and D are located on rockglacier No. 1). Note the increasing elevation between profiles A and D.**

Profile B (Figure 8; B) is located in hanging valley 3 on top of an old, terminal moraine crossing the stream, which drains the hanging valley (Figure 6). Surrounding dead ice holes indicate former subsurface ice occurrence behind the former moraine terminus. Complete vegetation cover of compresia pygmea interspersed with individual rockstones suggests an old and stable surface. From the high resistivity anomalies of up to 150 kΩm, we conclude that ice-poor permafrost in contrast to ice-rich permafrost in profile C and D is present as an ice lens at 5450 m a.s.l.

Profile C and D (possibly the highest-elevated ERT-measurements worldwide) show the typical two-layer structure of rockglacier No. 1 with equally high resistivity values (Figure 8; C, D). The first layer is characterized by lower resistivity values (1 – 20 kΩm), indicating the unfrozen active layer during the summer months. The active layer thickness varies between two and five meters. The second layer shows high resistivity values of up to 3500 kΩm and covers the complete section from below the active layer to the maximum depth of investigation. No internal heterogeneities are visible due to the lack of current flow within this highly resistive unit, which we interpret as a mixture of ice and sediment. According to Table 3, we interpret the second layer to be ice-rich permafrost. Similar resistivity values of ice-rich rockglacier material, reaching maximum values of 1000 kΩm have been reported in several studies from Häberli and Vonder Mühll (1996), Vanhala et al., (2009) and Mewes et al., (2017). Profile C and D confirm the presence of subsurface ice at an elevation around 5500 m a.s.l., which we use as evidence for the lower occurrence of probable permafrost.

The relatively large altitudinal steps between our four ERT profiles do not allow excluding the occurrence of subsurface ice in other, lower parts of the valley. Therefore, we use the following perennial creeping rates to exclude this case. The detection of subsurface ice is the second component of the three methods for estimating the probable occurrence of permafrost. Inferred by ERT-data subsurface ice can be expected at selected locations from an altitude of 5450 m and higher.

### 4.3 Creeping rates of periglacial landforms

The creeping rates for rockglaciers and protalus ramparts, including statistical information, are shown in Table 4. The fastest moving areas of landforms display lower coherence values and small spatial data gaps. The low coherence values in those areas are likely connected to the long temporal baselines of interferograms in summer of 2016 of up to 72 days and 96 days for ascending and descending data respectively. Long temporal baselines on relatively fast moving landforms may lead to aliasing effects if the displacement exceeds a quarter of the wavelength of the satellite (Crosetto et al., 2016). This would correspond to a LOS displacement of ~14 mm for Sentinel-1, which emits a wavelength of 56 mm. 14 mm in 72 days or 96 days corresponds to a LOS velocity of approximately 71 mm/yr for ascending and 53 mm/yr for descending data respectively. Displacement values in areas with higher LOS velocities than these thresholds are likely to be underestimated with the InSAR technique and display poor coherence values near or below the coherence threshold of 0.3. Coherence values do not drop significantly in winter, which is likely due to the semi-arid climate and therefore relatively thin snow cover.

**Table 4: Summary of InSAR- derived creeping rates for the periglacial landforms. The values represent the median of all data points over the entire observation period (2015-2018) on the respective landform. Uncertainty is given by the interquartile range in**

 **round brackets. The percentage of interpolated time periods describes how many interferograms are incoherent and therefore require interpolation with the ISBAS algorithm.**

| Landform | Creeping rate [mm/yr] | Summer acceleration [%] | Creeping rate precision [mm/yr] | Coherence | Interpolated time period [%] | Data points |
|---|---|---|---|---|---|---|
| **Protalus ramparts** | 11.0 (6.8 to 16.7) | -2 (-36 to 36) | 5.2 (3.9 to 7.8) | 0.74 (0.70 to 0.80) | 2.5 (0.7 to 6.2) | 7984 |
| **Rockglaciers** | 21.1 (11.6 to 36.8) | -23 (-46 to 3) | 5.1 (4.2 to 8.1) | 0.66 (0.59 to 0.73) | 7.9 (3.8 to 13.0) | 5402 |

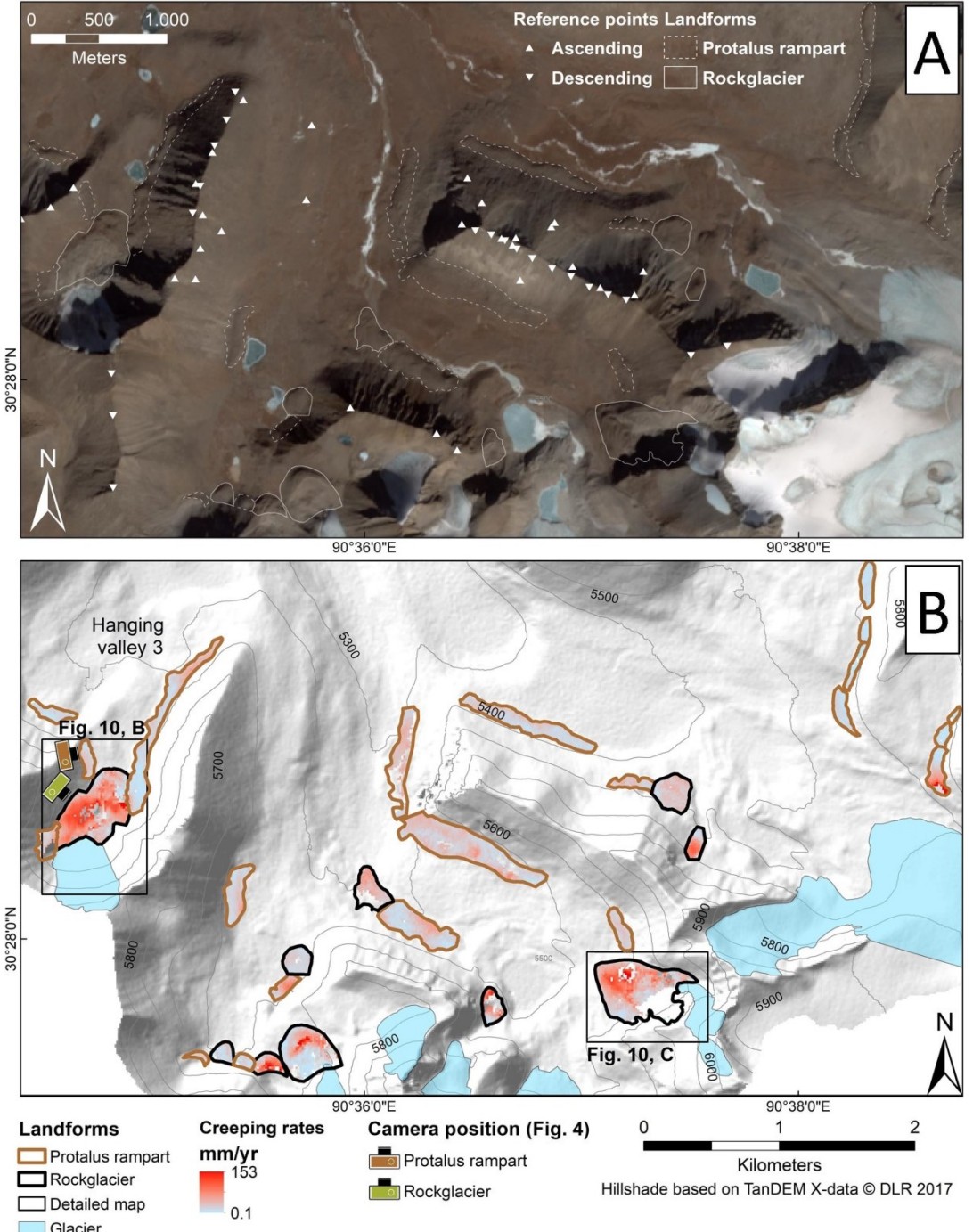

Figure 9: A: Sentinel II satellite image, recorded 30-01-2018. Triangles indicate stable reference points. Dashed lines indicate the outlines of the periglacial landforms. B: Creeping rates from periglacial landforms move in slope direction over the observation period 2015 - 2018. The Black rectangles mark the location of the two fastest rockglaciers in Figure 10. The Camera positions correspond to the Photographs in Figure 4.

Protalus ramparts in the Qugaqie Basin display lower average surface velocities than rockglaciers. The creeping rate of protalus ramparts (11.0 mm/y with an uncertainty from 6.8 to 16.7) is lower and shows more pronounced seasonal variations than on rockglaciers (21.1 mm/y with an uncertainty from 11.6 to 36.8). Rockglacier No. 1 of hanging valley 3 which we also studied with ERT measurement displays creeping rates of up to 70 mm/yr in most areas, with the fastest moving part reaching 153 mm/yr (Figure 10, B), similar to the rockglacier No. 2 (Figure 10, C). A time series of creeping rates of the rockglacier No. 1 is shown in Figure 10 A (black line) and of the rockglacier No. 2 in Figure 10 A (grey line). The spatial distribution of the creeping rates is relatively uniform in areas with good InSAR sensitivity, i.e. slopes with an East or West aspect, but displays significantly higher noise level in areas with poor InSAR sensitivity, i.e. slopes with a North or South aspect.

We do not observe a clear correlation between variations in creeping rates and possible seasonal forcing mechanisms such as temperature or precipitation. Neither protalus ramparts nor rockglaciers display clear acceleration of creeping in summer compared to winter (Figure 10, A).

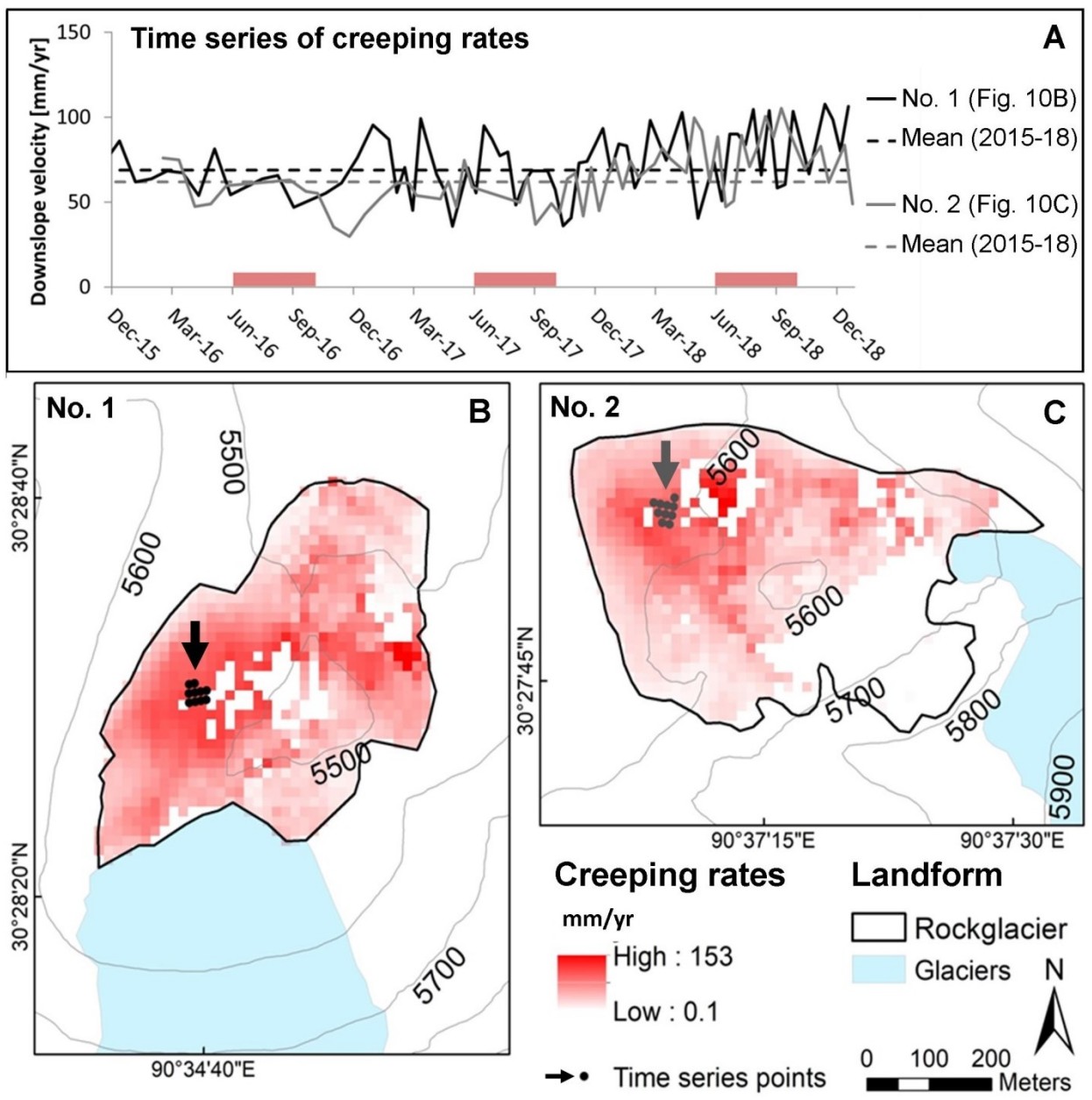

Figure 10: A: Summer months with air temperatures >0 °C (according to Zhang et al., 2013) are shown in red. Time series represent the moving average of the 10 nearest values in time based on the median of time series points, located in B (black dots) and C (grey dots). Black time series (rockglacier No. 1 in B) is based on ascending data and grey time series (rockglacier No. 2 in C) on descending data.

The third component for assessing the occurrence of permafrost is based on the movement rates of periglacial landforms. Based on the assumption that a measurable movement rate is determined by perennial ice in the subsurface, the observed active status of the periglacial landforms allows the conclusion of permafrost occurrence in the corresponding landform.

## 4.4 Assessment of the lower permafrost limit of the Qugaqie valley

The assessment of the lower permafrost limit consists of an integration of different results. The field based mapping of periglacial landforms indicates the first precondition to find permafrost conditions. Field observations like furrows, ridges, coarse substrate and lichen coverage on the rockglaciers surface corroborate the mapped landforms classification and indicate activity of the landform. By integrating the ERT-results of detected subsurface ice occurrence a further component of the permafrost condition (subsurface below 0 degree) is validated. Completing the permafrost definition (of two or more consecutive years) the derived creeping rates by InSAR show a constant motion of more than two years which is attributed to the deformation of the debris ice matrix of the periglacial landforms. So, the active status, the altitudinal distribution of the periglacial landforms and validated ice-occurrence by ERT suggest a lower limit of probable permafrost between 5300 – 5450 m a.s.l. This range includes ice lenses detected by ERT-data as well as all creeping landforms indicating an active status and therefore an existence of ice.

## 5 Discussion

One critical issue for the estimation of the lower occurrence of probable permafrost by the used approach is the focus on periglacial landforms. These landforms are characterized by blocky material and a special thermal regime that lowers the internal temperature in comparison to the thermal regime outside of the blocky, rough surface (Gorbunov et al., 2004). This cooling effect of high-porosity unconsolidated debris is especially observed in lower mountain regions by near-surface ground temperature measurements on rockglaciers (Onaca et al., 2020) and suggests a lowering of discontinuous and sporadic permafrost occurrence (Lambiel and Pieracci, 2008; Otto et al., 2012). By using the ERT-method we found ice-poor permafrost in ice-lenses in mineral soils next to the rockglacier that corroborates the idea of permafrost conditions outside of blocky material at an elevation of 5450 m a.s.l. The extreme cold mean annual air temperature of -6.8°C at 5680 m a.s.l. (Zhang et al., 2013) should minimize the effect of different regolith properties that favors permafrost conditions.

The next critical issue for the estimation of the lower occurrence of probable permafrost is the question whether the huge resistivities observed on profile A (Figure 8, A, black lines) indicates ice or not. In general, subsurface material determination without additional cross-validating techniques by other geophysical methods or borehole data remains uncertain (Hauck and Kneisel, 2008; Guglielmin et al., 2018). Therefore, the geomorphological knowledge of the study area

is essential for an interpretation of the subsurface: In this case, the measured resistivity values of Profile A (Figure 8) of up to 100 kΩm are consistent with both till or ice-poor permafrost (Schrott and Sass, 2008). From the resistivity values it is therefore not possible to determine whether the till contains ice or not. However, field observations allow the conclusion that no ice was measured because clear creep structures would have to be recognizable due to a significant slope. Furthermore, InSAR analysis of this location shows no clear perennial creep behaviour (Reinosch et al., 2020), making the presence of subsurface ice unlikely. In order to uniquely identify ice, it would have been desirable to apply additional, geophysical methods, like ground penetrating radar, refraction seismic tomography, or capacitively coupled resistivity (Mudler et al., 2019). In particular, the combination of electrical and seismic methods allows to derive a petrophysical four-phase model (Hauck et al., 2008; Mewes et al., 2017) and to estimate the sediment-to-ice ratios from electrical resistivity and seismic velocities. However, due to the extremely difficult logistical constraints in this remote location, these methods could not be applied, and we thus rely on combining evidence from field observations with geophysical results.

The approach by Kneisel and Kääb (2007) uses a similar combination of methods as used in this study to describe periglacial morphodynamics of a glacier forefield including a rockglacier. ERT profiles show the same range of layer thickness of 2-5 meters as in our profiles in the summer months. They recommend the joint application of geoelectrical and surface-movement data to investigate periglacial landforms and to assess the permafrost distribution, because the combination of both tools allows a more comprehensive characterization of permafrost characteristics like ice-rich or ice-poor. Also, in our case, we believe the ground-based geophysical surveys are useful, as predicting subsurface ice content and deriving permafrost distribution maps only by modelling and/or using remote sensing includes various sources of error:

- Low resolution (1 km gridded) of the permafrost-distribution models over the entire TP (Zou et al., 2017; Figure 1, C) prevents detailed analyses of permafrost occurrence at a meso-scale, especially in high mountain relief.
- Surface displacement patterns originate from different surface processes and take place in different time-intervals, such as freeze-thaw cycles, seasonal creeping or constant, multiannual creep (Reinosch et al., 2020).
- Remote sensing approaches can only guess the geomorphological process behind the surface displacement. Surrounding landscape features, underlying material and sediment source areas are essential factors that need to be considered during the interpretation of remote sensing imagery.
- Without ground based validation (e.g. ERT-data) large-scaled permafrost distribution maps cannot accurately be used to predict permafrost occurrence in the remote, high mountain areas.

Geomorphological field evidence allows a small-scaled interpretation and, in combination with remote sensing data an extrapolation to larger scales. The periglacial landforms in this study show lower creeping rates than similar landforms of other regions. Other studies employing InSAR techniques observe creeping rates from centimetres to several meters per year for rockglaciers in the western Swiss Alps (Strozzi et al., 2020), in western Greenland (Strozzi et al., 2020) and in the Argentinian Andes (Villarroel et al., 2018; Strozzi et al., 2020). Furthermore, all of them clearly indicate seasonal variations

of the rockglacier movement, with faster rates in summer and reduced creeping rates in winter months (Cicoira et al., 2019; Delaloye et al., 2008, 2010). In our study area neither rockglaciers nor protalus ramparts display significantly accelerated creep in summer (Figure 10, A). The lack of seasonality and the lower creeping rates compared to rockglaciers in the alps (Cicoira et al., 2019; Kenner et al., 2017; Wirz et al., 2016) and the semi-arid Andes (Strozzi et al., 2020) might be related to the semi-arid climate conditions (lack of moisture) and the short time-span of three months with positive air temperatures in the Qugaquie basin (Zhang et al., 2013). Strozzi et al. (2020) figured out that their highest rockglacier "Dos Lenguas" (4300 m a.s.l.) in the Andes is characterized by "less amplitude variations of the annual cycle than observed for the Swiss Alps". Hence, we hypothesize that the seasonality of rock glacier creeping behaviour is less pronounced the lower the mean annual air temperatures and the shorter the timespans of positive air temperature are. It seems that the magnitude of seasonal variations of the creeping rates also decreases with a lower availability of moisture, because strongest seasonality is observed in moist regions such as the Alps. Additionally, catchments in the Qugaquie basin are quite small for sediment release, so the extent of our rockglaciers is limited by a small debris input. Probably for similar reasons, protalus ramparts investigated in this study creep with an median velocity of 11 mm/y, while comparable creeping rates for protalus ramparts range from 40 cm/y up to 100 cm/y in the Swiss alps (e.g., Scapozza, 2015).

The optical image-based process of rockglacier mapping and outlining is subject to several uncertainties, like the quality of optical imagery and the rather subjective mapping style (Brardinoni et al., 2019). However, rockglacier inventories become increasingly important due to their function as indicators of stored water resources (Azócar and Brenning, 2010; Jones et al., 2018b, 2018a) and their response to climate (Cicoira et al., 2019; Humlum, 1998). An IPA-working group was installed to reduce the uncertainties of such inventories and to standardize mapping procedures (Delaloye et al., 2018). This year (2020) standardized guidelines were published on: ([https://www3.unifr.ch/geo/geomorphology/en/research/ipa-action-group-rock-glacier/](https://www3.unifr.ch/geo/geomorphology/en/research/ipa-action-group-rock-glacier/)) which we followed in our mapping procedure (Delaloye and Echelard, 2020). Additionally with the opportunity to perform a field-based mapping a decrease of these uncertainties is likely.

Using rockglaciers and their long-term ice content as indicators for permafrost occurrence must be critically evaluated because rockglaciers can overcome long distances and the terminus is far away from the routing zone  (Bolch and Gorbunov, 2014). In this case rockglaciers are not suited for permafrost distribution assessment, because the ice-debris mass creeps out of the continuous permafrost zone, as the rockglacier distribution in combination with modelled permafrost occurrence demonstrates in the northern Tien Shan (Bolch and Gorbunov, 2014). In our study, periglacial landforms are characterized by a small extent and a low altitudinal range in extreme elevation. The rockglacier terminus is close to the rooting zone, and they do not span a significant elevation range. Temperature data (MAAT of – 6,8°), elevated at Zhadang glacier (Zhang et al., 2013), and different, large scaled permafrost distribution maps (Zou et al., 2017, Obu et al., 2019) suggest a high permafrost probability at elevations greater than 5400 m a.s.l. in the study area. Nevertheless a detailed, small scaled model of permafrost distribution would help to make a prognosis of permafrost occurrence by localizing probabilities especially in

lower areas of the catchment. "Permakart" considers topographic parameters and different slope characteristics by using a topo-climatic key to handle the heterogeneity of high mountain areas (Schrott et al., 2012).

**6 Conclusion and future work**

In spite of the adverse logistical conditions in the study area, we were able to give insights to the cryosphere and to assess a lower permafrost occurrence in the Qugaqie Basin on the TP using a multi-method approach. Thus, we add an important

piece of information to the literature in a region where, due to its high altitude, ground truth data is usually difficult to obtain. Geomorphological mapping identifies the altitudinal distribution of periglacial landforms. ERT-measurements validate ice occurrence of one periglacial landform, a rockglacier. The activity of the periglacial landforms is derived from surface displacement analysis of high resolution InSAR-data over three years. By combining the three findings we assess the lower occurrence of probable permafrost. The main outcome is summarized as follows:

- The altitudinal distribution of periglacial landforms ranges between 5300 m a.s.l. and 5600 m a.s.l. and averages around 5500 m a.s.l. Protalus ramparts are more frequent, while rockglaciers have a larger extent and creep faster.
- ERT measurements outside of blocky material of the periglacial landforms indicate ice-poor permafrost such as ice-lenses (70 – 150 k$\Omega$m) at 5450 m a.s.l.
- ERT measurements on a rockglacier confirm perennial ice occurrence around 5500 m a.s.l. Resistivity values of more

than 200 k$\Omega$m indicate ice-rich permafrost.
- Surface displacement analysis extrapolates the status of active creeping to other permafrost related landforms. Especially rockglaciers show creeping rates up to a maximum of 150 mm/y (median 21 mm/y). Protalus ramparts have much lower surface creeping rates (median 11 mm/y).
- Seasonality of rockglacier creep is lacking probably due to low average temperatures and semi-arid climate

conditions.
- The lower limit of probable occurrence of permafrost is higher than 5300 – 5450 m. a.s.l.

Our results illustrate the benefit of combining field-based and remote sensing techniques and recommend interdisciplinary approaches to geomorphological and geocryological issues. Nevertheless, the current results should be compared with a permafrost model of the study area in order to make a prognosis and zonation of the permafrost distribution. We also follow

the suggestion by Strozzi et al. (2020) to include rock glaciers and the monitoring of rock glacier velocities as an essential climate variable in the Global Climate Observing System (GCOS) of the World Meteorological Organization due to the essential contribution of the results as climate sensitive parameters. As a next step, we plan to provide a rock glacier inventory for the western Nyainqêntanglha Range based on InSAR-data as a status quo to understand the sensitivity and the vulnerability of high mountain cryosphere referred to climate warming.

*Data availability.* The data sets can be obtained on request to the authors.

*Team list and Author contributions.* JB designed the study, conducted fieldwork, processed and interpreted geomorphological and geophysical data, wrote the manuscript and conceptualized figures. ER was in charge of InSAR analyses. AH participated in fieldwork, helped with the ERT data processing and revised the manuscript carefully several times. BR conducted field logistics and data acquisition. MG and BR participated in the interpretation of the remote sensing data. FZ provided substantial logistical support and contributed to the discussions on the data interpretation. AS, FZ and RM actively participated in the data interpretation and worked out the scientific concept underlying the research proposal leading to this work. All authors contributed to the revision of the text.

The authors declare that they have no conflict of interest.

**Acknowledgements**

We thank all colleagues who contributed to this study, especially Bernd Wünnemann for fruitful discussions and valuable comments during field work, Jussi Baade for providing the TanDEM-X data (©DLR), Zhengliang Yu for the support in the field and Guoshuai Zhang and his team for logistical support at the Nam Co Station for Multisphere Observation and Research, Chinese Academy of Sciences. We thank Matthias Bücker and Felix Nieberding for the insightful and highly valuable suggestions during this manuscript's preparation and revision phase.

We thank the editor Tobias Bolch and two anonymous reviewers for their comprehensive work on comments and suggestions to improve the manuscript.

This research is a contribution to the International Research Training Group (GRK 2309/1) "Geo-ecosystems in transition on the Tibetan Plateau (TransTiP)" funded by Deutsche Forschungsgemeinschaft (DFG).

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
