# Peer review of "Insights in a remote cryosphere: A multi method approach to assess permafrost occurrence at the Qugaqie basin, western Nyaingêntanglha Range, Tibetan Plateau"

_The Cryosphere, 2020_

## Referee Comment (RC1) · Anonymous Referee #1 · 27 Jun 2020

Review of "Insights in a remote cryosphere: A multi method approach to assess permafrost occurrence at the Qugaqie basin, western Nyainqêntanglha Range, Tibetan Plateau" by Johannes Buckel et al.

The authors integrated three distinct methods, including geomorphological mapping, geophysical (ERT) survey, and radar remote sensing, to infer the lower altitudinal limit of permafrost based on periglacial landforms and their surface movements in the remote Qugaqie basin in Tibet. As well-dressed in this manuscript, each method suffers its intrinsic limitations; and their combination makes a strong and convincing case towards a quantitative, detailed, and comprehensive assessment of permafrost occurrence within a basin where no borehole measurements are available. In addition to this success in method integration, I would also commend the authors' efforts of conducting field mapping and geophysical surveys in this remote and harsh environment. Overall, periglacial geomorphology in Tibet is poorly studied. This work provides a valuable set of observations and datasets, which hopefully will be published and help to generate more interest in studying geomorphology in this area.

However, I would raise a few issues regarding the scientific meanings, the added value of combining three methods, some details of methodology, and clarity in writing. I also provide a long but incomplete list of editorial comments. A careful proofread is needed.

1. Scientific meanings. Even though three research questions are listed in the introduction, it is unclear to me why the authors wanted to assess permafrost occurrence in this particular basin and why we should care. Moreover, what are the implications of the inferred altitude limit of around 5400 m. And how does the limit in this basin compared with other places in Tibet (this comparison is hinted, but not explicitly addressed). Lastly, please define what do you mean by 'probable permafrost'? Why and what part of your assessment is 'probable', not deterministic?

2. The authors explained complementary nature of the three methods very well in the introduction section and beginning of section 3 with figure 2. My question remains in the quantitative interpretation of the elevation (or elevation ranges) as inferred from each method (e.g., lines 26-28 in the abstract, and summarized in the first three bullet points in the conclusion section). Here is a list of related sub-questions.

2a. What was the exact reason to give one single value (namely 5400 m) for the lower limit of 'probable permafrost'? What about the two protalus ramparts mapped that are located between 5300 m to 5400 m? Hypothetically, if one doesn't conduct ERT or InSAR measurements, would it be still reasonable to give an estimate of 5300 m with some uncertainty given?

2b. Quantitively, what are the added value of the ERT results? Are these localized site surveys critical or supplementary for arriving the 5300 m estimate?

2c. As the InSAR measurements only provide information on surface movement, am I right to interpret that InSAR is marginally supplementary for estimating permafrost occurrence in this study? And it is not very clear what the authors mean by InSAR "allows a **transfer** and an **extrapolation** of our findings about ice occurrence from geophysical measurements to other

periglacial landforms" (Line 84). How exactly did you transfer and extrapolate using the InSAR-measured surface movement?

3a. I wish the authors could provide more details of the geomorphological mapping in section 3.1. In particular, how did you distinguish periglacial landforms that are characteristics of seasonally-frozen ground and permafrost (Line 168)? How did you make use of the DEM and optical imagery to map rock glaciers and protalus ramparts, or were they mapped mainly based on field observations? It might be useful to show optical images/DEM over some of the rock glaciers and protalus ramparts (e.g., next to figure 6 or 9, or as background for figure 10). Why did you need to use optical images from various sources? What are their spatial resolutions? BTW, the Digital Globe, BING, and Google Earth images were mostly likely taken by satellites, therefore not aerial images (Line 165). Is it possible that some landforms were missed and not included in the geomorphological mapping?

3b. It would be helpful if the authors could add a brief description of the roll-along procedure illustrated in figure 5. Table 2 and associated text fit better in section 3.2.

3c. The authors first reported low InSAR coherence in spring and summer (line 239), but later stated as in spring and autumn (line 270). Did I miss anything? And could the authors add some description of the intermittent SBAS approach? Could you also include velocity ranges in Table 3?

Figure 1: I cannot find blue, red, or black arrows in 1A. Label "Namco" should be "Nam Co"

Figure 6: The black rectangle is labeled as Figure 10, but should be Figure 9. The colors for lakes and glaciers are very similar (can set a darker color for lakes). It would be ideal to use an identical set of colors for the landforms in both figures 6 and 9.

Figure 9: (this is a minor visualization issue). InSAR-measured creeping rates show as light yellow in some landforms (e..g the two lowest ramparts and in the middle of some rock glaciers), which are not well distinguishable from the light-yellow elevation mask.

**Editorial comments:**

No need to capitalize permafrost if it is not at the beginning of a sentence.

L17: play should be plays

L19: at should be in

L50: decrease should be decreased

L55: spell out ERT here, at its first appearance in the text

L56: part should be parts

L61 & L69: at the TP should be on the TP

L79: which should be what

L93: cretaceous should be Cretaceous

L96: In should be On

L97: spell out ISM here at its first appearance in the text

L118: 'the' GLIMS database (can also spell out GLIMS)

L120: then should be than

L128: two-years should be two-year

L152: "derived the from the mean"?

L190: is 'pronival rampart' interchangeable with protalus rampart? If so, it is better to stick with protalus rampart.

L235: changes should be change

L328: "in terms their interpretation in terms of material characteristics"?

L405: are should be is

L408: regimes should be regime

L409: effect should be effects

L410: an should be and?

L410: cower should be cover

L411: delete one 'at'

L412: cause should be because

L418: method should be methods

L431: delete 'the' before this

L466: of should be away

L466: routing should be rooting?

L469: occurence should be occurrence

L472: scalled should be scaled (or large-scale)

L473: probabilities should be probability

L478: insight is a noun

L483: asses should be assess

---

## Referee Comment (RC2) · Anonymous Referee #2 · 29 Jul 2020

Review of "Insights in a remote cryosphere: A multi method approach to assess permafrost occurrence at the Qugaqie basin, western Nyainqêntanglha Range, Tibetan Plateau" by Johannes Buckel et al..

In this interesting work, the authors combined three distinct methods: (i) geomorphological mapping to identify periglacial landforms, (ii) geophysical (ERT) survey to evaluate the presence of ice, and (iii) satellite Synthetic Aperture Radar Interferometry to investigate surface displacements of periglacial landforms that can be used as a proxy of ice occurrence. These methods were applied in the remote Qugaqie basin in Tibet. Despite the logistical problems due to high altitude and extreme climatic conditions, field campaigns were conducted in this area for ERT survey. This study increases the knowledge of poorly studied periglacial geomorphology in Tibet, including the permafrost occurrence within a basin where no borehole measurements are available.

Here some general comments about the manuscript, followed by some detailed comments. A careful proofread is needed.

General comments.

The study area and the three methods used in this work are well described in their respective sections. However, further information on the approaches and criteria used to outline the identified landforms should be added, including some examples with their associated uncertainties.

About the method that describes the Synthetic Aperture Radar Interferometry, different geometries (i.e. ascending and descending) are used. However, the Authors should explain how they used these different geometries (combination of both, choosing the most appropriate, etc.).

In the results section, velocity values are showed in the text, tables and figures. However, disagreements are apparently visible, probably due to the different "observation periods" in which the velocities were computed (e.g. velocity computed in annual periods vs velocity computed in summer periods?) not specified in the text. The Authors should clarify how the velocity are computed and in which observation periods, in order to avoid misunderstandings. Furthermore, measurements are often shown with different units, but the same unit should be used (mm/yr and mm or cm/yr and cm).

In the discussions there is not a paragraph or section that discusses the mapping of landforms. The same landform can be outlined following different approaches. As suggested by Brardinoni et al. 2019 (https://doi.org/10.1002/esp.4674) the uncertainty on mapping rock glacier is very high. IPA Action Group is now working on baseline concepts and practical guidelines to produce uniform rock glacier inventories (https://www3.unifr.ch/geo/geomorphology/en/research/ipa-action-group-rock-glacier/).

Detailed comments

L 56: unclear sentence.

L 78: this sentence seems to be a "general research question". As you already mention, Barsch, 1996; Schrott, 1996, Scapozza, 2015 say this. Specify that this sentence is related to TB landforms.

L 97: specify what the ISM acronym is (now it is in the L 106).

L 114, Figure 1A: wind systems are not visible in the figure A. Add the position of Nyainqêntanglha in figure A.

L 152 – 154: unclear sentence.

L 168: here you say "excluding glacially-conditioned environments", but in the following parts you mapped also glaciers. Be clearer.

L 171: the meaning of "manually" is unclear (after the field campaign?). Be clearer.

L 173: add the parameters used to derive the hillshade (azimuth and inclination of light source).

L 175: IPA Action Group is working on Baselines and Practical Guidelines for mapping rock glaciers.

L 176: unclear sentence.

L 181: Location of the photos at Figure 6 and 9.

L 236: specify that displacements higher than half of the SAR wavelength are undetectable (e.g. rockfall) and coherence also decreases.

L 237: How do you choose the threshold of 0.3? Motivate your choice.

L 254: Describe how you used the ascending and descending dataset (do you combine them? Or do you select one depending on the landform orientation?).

L 262: coefficient lower than 0.2 is possible, but it is not used in order to avoid an excessive amplification of displacements and associated errors (from Notti et al. 2014).

L 274: a table containing the parameters used during the processing (or the reference) can be useful here.

L 277: provide the locations and outlines of these stable areas (e.g. in figure 6 or 9).

L 300, figure 6: Inside the figure 6 replace Figure 10 with 9.

L 312: talus slope and protalus rampart also provide sediments in the rooting zone of rock glaciers (visible in figure 6).

L 346, figure 8: use the "k ohm m" unit in the figure 8 (as in the text). Add the cartesian coordinates on each profile (e.g. West and East for A A').

L 365: specify how you compute the mean velocity of the landforms, not only in the caption of table 3 (do you selected some points inside the landforms? Or do you use all the points inside the delineated landforms?), and add the observation periods (e.g. velocity computed in annual periods or velocity computed summer periods?).

L 371: use always the same units in the text (mm/yr and mm or cm/yr and cm).

L 376, table 3:

- the computation of "downslope velocity precision" is not clear.
- the "interpolated time periods" is not defined.
- there are apparent disagreements between this table and the text below. For example, from this table the maximum downslope velocity expected for rock glacier is 47.2 mm/yr (26.8±20.4), but velocities up to 153 mm/yr are visible in figure 9 and 10. This is probably due to different points investigated or different "observation periods" where the mean velocities were computed (e.g. annual periods vs summer periods?). please specify how these values are obtained.

L 386, Figure 9: Specify if the "creeping rates" are along LOS or slope direction. Indicate the observation period of the velocities showed in the figure.

L 390: these values ("from 10 mm/yr to 80 mm/yr") are apparently in disagreement with those in table 3 (the mean downslope velocity is 12.7±9.2 mm/yr). Explain why (different observation periods?).

L 397 – 398: this sentence is not supported by results and data. Time series of temperature and precipitation are not shown. Provide data (or references) to support, or remove this sentence.

L 401, figure 10: indicate the observation periods of the velocities showed in the maps B and C.

L 433: specifies whether the layer refers to the ice layer.

L 466 – 469: rephrase these two sentences (it is not clear that these two sentences are not related to your work).

L 483: Sentinel 1 is not a high-resolution satellite.

---

## Author Comment (AC1) · 13 Aug 2020

This document includes the author's comments *(italic font)* to the review (normal font) RC1 of Anonymous Referee #1, from 27 Jun 2020. Planned changes in the revised documente are mentioned in […].

Review of "Insights in a remote cryosphere: A multi method approach to assess permafrost occurrence at the Qugaqie basin, western Nyaingéntanglha Range, Tibetan Plateau" by Johannes Buckel et al.

The authors integrated three distinct methods, including geomorphological mapping, geophysical (ERT) survey, and radar remote sensing to infer the lower altitudinal limit of permafrost based on periglacial landforms their surface movements in the remote Qugaqie basin in Tibet. As well-dressed in this manuscript, each method suffers its intrinsic limitations and their combination makes a strong and convincing case towards a quantitative, detailed, and comprehensive assessment of permafrost occurrence within a basin where no borehole measurements are available. In addition to this success in method integration, I would also commend the author's efforts of conducting field mapping and geophysical surveys in this remote and harsh environment. Overall, periglacial geomorphology in Tibet is poorly studied. This work provides a valuable set of observations and datasets which hopefully will be published and help to generate more interest in studying geomorphology in this area.

However, I would raise a few issues regarding the scientific meanings, the added value of combining three methods, some details of methodology, and clarity in writing. I also provide a long but incomplete list of editorial comments. A careful proofread is needed.

1. Scientific meanings. Even though three research questions are listed in the introduction, it is unclear to me why the authors wanted to assess permafrost occurrence in this particular basin and why we should care. Moreover, what are the implications of the inferred altitude limit of around 5400 m. And how does the limit in this basin compared with other places in Tibet (this comparison is hinted, but not explicitly addressed). Lastly, please define what do you mean by 'probable permafrost'? Why and what part of your assessment is 'probable', not deterministic?

*Authors comment: This particular basin is used for permafrost research because of its high elevation and the location in a mountain range at the TP. In general, Permafrost research at the TP is strongly connected to infrastructure and anthropogenic implications. Aside from this, permafrost-related data is lacking, especially in mountainous regions. We will modify the introduction to stress these aspects more clearly.*
*We will add in L 67: [Large scale modelling of PF-conditions on the TP (Sun et al., 2020) show that the study area (Figure 1, B) is located at the interface between continuous permafrost and seasonally frozen ground (Figure 1, C), which makes it a suitable environment to validate such large-scale models and to precise the interface with ground-truthed data. The validation is important, because the final conclusion would be that the TP is not completely underlying permafrost conditions, unlike expected and modelled at other places in Tibet (Cao et al., 2019; Ran et al., 2012)].*

*We agree that the term "probable permafrost" is not well defined, and that the lack of a precise definition might cause confusion. Therefore, we change the term to "probable occurrence of permafrost". The use of the term "probable" is motivated by the fact that we do not have borehole-data (MAGST) and a modelled permafrost distribution and we can only assess the occurrence indirectly. The spatial*

*heterogeneity of our data (mapping, InSar and ERT) and of natural variations in permafrost occurence also prevents us from providing precise limits.*

*We will change in L 67-69: [. This study aims to supplement the previously summarized studies with an assessment of probable occurrence of permafrost in remote high mountain regions away from the Tibetan engineering corridors and to provide a ground truthing for existing permafrost studies and maps at the TP. The use of the term "probable" is motivated by the fact that we do not have borehole-data (Mean annual ground surface temperature) and no modelled permafrost distribution is available. So, we can only assess the occurrence indirectly. The spatial heterogeneity of our data (mapping, InSar and ERT) and of natural variations in permafrost occurrence also prevents us from providing precise elevational limits, so we provide an assessment of probable occurrence of permafrost in a range according to the findings of the three methods]*

*We will change in L 87: [As an assessment, the study provides probable occurrence of permafrost by combining these three methods for a catchment in an high-altitude mountain range of the TP.]*

2. The authors explained complementary nature of the three methods very well in the introduction section and beginning of section 3 with figure 2. My question remains in the quantitative interpretation of the elevation (or elevation ranges) as inferred from each method (e.g., lines 26-28 in the abstract and summarized in the first three bullet points in the conclusion section). Here is a list of related sub-questions.

*We answer the reviewer's remaining question in the final manuscript as follows. Each result section of the corresponding methods will receive a concluding sentence indicating the inferred elevation of probable permafrost occurrence.*

*We will add in L 324: [The meso-scaled periglacial landforms (mean elevation) are situated between 5300 and 5600 m a.s.l. This altitudinal distribution serves as one component of the three methods for assessing the probable occurrence of permafrost in the catchment.]*

*We will add in L 364: [The detection of subsurface ice is the second component of the three methods for estimating the probable occurrence of permafrost. Inferred by ERT-data subsurface ice can be expected at selected locations from an altitude of 5450 m and higher].*

*We will add in L 403: [The third component for assessing the occurrence of permafrost is based on the movement rates of periglacial landforms. Based on the assumption that a measurable movement rate is determined by perennial ice in the subsurface, an active status allows the conclusion of permafrost occurrence in the corresponding landform.*

2a. What was the exact reason to give one single value (namely 5400 m) for the lower limit of 'probable permafrost'? What about the two protalus ramparts mapped that are located between 5300 m to 5400 m?

*One aspect of our study is that it provides localised ground-truth data in an area that is otherwise only accessible by simulation or theoretical considerations within large-scale models. Therefore, in order to compare the result with existing literature, we find it useful to infer one single value. In reality, the occurrence of permafrost naturally varies depending on topo-climatic factors. To take this into account, the single value is transformed into a range, which also includes the two lowest, active protalus ramparts.*

*We state out as a general outcome of the publication "the probable occurrence permafrost is higher than 5300 - 5450 m. a.s.l"*

*The following Changes will be made:*

*L 25: [ to assess the probable occurrence permafrost in the Qugaqie basin]*

*L 31: [ probable]*

*L 70:  The identification of periglacial landforms, subsurface ice and surface creeping rates on these landforms leads to an assessment of the probable occurrence of permafrost.*

*L 306: [We conclude from this altitudinal distribution  a probable occurrence of permafrost higher than 5300 m.a.s.l., which has to be supported by validating ice occurrence and the status of activity of these landforms.]*

*L 493:[ The probable occurrence of permafrost ranges higher than 5300 – 5450 m. a.s.l.]*

2b. Quantitively, what are the added values of the ERT results? Are these localized site surveys critical or supplementary for arriving the 5300 m estimate?

*ERT is a common method to detect ground-ice in the subsurface, inferring permafrost conditions (Lewkowicz et al., 2011), if ground ice is present for two consecutive years.  Therefore, ERT supplements InSar-findings and vice versa. Surface creep without ice could happen without permafrost conditions and subsurface ice without creep over 2 consecutive years is not related to permafrost. With the help of ERT we were able to provide evidence for the existence of ground ice at specific test sites. The localized site surveys are critical for arriving at the 5300 – 5450 m estimate. Profile A (Fig. 8) ranges from 5090 to 5230m and no subsurface ice content was detected in the lateral moraine, which could be supposed.*

*We will add in L 325: [ERT is a common method to detect ground-ice in the subsurface, inferring permafrost conditions (Lewkowicz et al., 2011), if ground ice is present for two consecutive years. With the help of ERT we were able to provide evidence for the existence of ground ice at specific test sites. The localized site surveys are critical for arriving at the 5300 – 5450 m estimate because no high resistivity values representing subsurface ice content were measured. Profile A (Fig. 8) ranges from 5090 to 5230m and represents subsurface conditions in the lower altitudinal areas of the catchment, for example in a lateral moraine.]*

2c. As the InSAR measurements only provide information on surface movement, am I right to interpret that InSAR is marginally supplementary for estimating permafrost occurrence in this study? And it is not very clear what the authors mean by InSAR "allows a transfer and an extrapolation of our findings about ice occurrence from geophysical measurements to other periglacialcial landformsm (Line 84). How exactly did you transfer and extrapolate using the InSAR-measured surface movement?

*Surface movement is one key parameter of periglacial landscapes (Eckerstorfer et al., 2018) and is used to infer permafrost conditions (Kneisel and Kääb, 2007; Strozzi et al., 2004). Therefore, InSar-data plays*

*a major role in this study, because this data validates the two-years rule of frozen ground (=permafrost) through the creeping behavior (compare Fig. 10A).*
*We will delete the thought of transfer (line 84) because of possible confusion.*

*We will add in L.152 [Although the continuous movement of periglacial landforms and the presence of ice can be implied from InSar data alone, ground truth at selected locations by ERT is essential to exclude other possible explanations.]*

3a. I wish the authors could provide more details of the geomorphological mapping section 3.1. In particular, how did you distinguish periglacial landforms that are characteristics of seasonally-frozen ground and permafrost (Line 168)?

*A differentiation of seasonal and perennial frozen ground of the study area is given by Reinosch et al. (2020). Based on InSar time series analyses of the typical periglacial surface changes the authors developed a model to distinguish annual surface changes (seasonal frozen ground) from perennial, more or less constant creeping surface changes (perennial frozen ground) for the study area. We focus directly on periglacial landforms according to perennial creep caused by down moving ice content.*

*To clarify this issue, we add in L 169: [A differentiation from seasonally-frozen and perennially-frozen movement behavior is given by the data and a derived model by Reinosch et al. (2020). This data was provided for the preparation for the periglacial landform inventory.]*

How did you make use of the DEM and optical imagery to map rock glaciers and protalus ramparts, or were they mapped mainly based on field observations?
*This question will be answered below the next question.*

Why did you need to use optical images from various sources? What are their spatial resolutions? BTW, the Digital Globe, BING, and Google Earth images were mostly likely taken by satellites, therefore not aerial images (Line 165). Is it possible that some landforms were missed and not included in the geomorphological mapping?

*The periglacial landforms were mapped at first mainly based on field observations. Back at the office the digitizing procedure started. Therefore, different optical imagery was used because of differing cloud and snow cover to ensure identifying periglacial features clearly. A slope and a hillshade map help to identify landform features more accurately.*
*To map smaller landforms more time would be needed. Therefore, smaller landforms could be missed, instead the meso-scalled landforms in focus are completely mapped. The DEM originates from TanDEM-X data (2015) with a resolution of 12 m (©DLR). The optical imagery based on BING maps (2013, 15m resolution of TerraColor imagery and 2.5m SPOT Imagery) and Google Earth data (2007-2012) provided by digital globe in a resolution between 0.5 and 5 m resolution depending on the satellite. We will add more detailed information in the revised manuscript.*

It might be useful to show optical images/DEMs/DEM over some of the rock glaciers and protalus ramparts (e.g., next to figure 6 or 9, or as background for figure 10).

*In the revised version, we will split Figure 9 in Part A and B. Both Parts show the same map extent. Part A displays a Landsat 8 image with the focus on the image, B focusses on the features and their creeping rates.*

[Figure]

*In Line 385 the following text will be added: [Figure 9, A: Landsat 8 satellite image, recorded 30-01-2018. Triangles indicate stable reference points. Dashed lines indicate the outlines of the periglacial*

*landforms. B: Creeping rates from periglacial landforms move in slope direction over the observation period 2015 - 2018. Black rectangles display the two fastest rockglaciers in Figure 10. Camera position shows the Photographs in Figure 4*

3b. It would be helpful if the authors could add a brief description of the roll-along procedure illustrated in figure 5..

*In Line 210 will be added: [For this procedure, two cables were available (denoted A and B), each equipped with 50 channels. First, both are connected with the control unit to obtain pseudosection number 1 (Figure 5). Next, cable B (and all connected electrodes) remains at the same location, whereas cable A is moved to the right of cable B to measure the Preudosection number 2, and so on.]*

Table 2 and associated text fit better in section 3.2.

*The location of table 2 is probably debatable, but since the compiled resistivity values are results and are needed for the explanation of the ERT results, we would prefer to leave the table in section 4.2. We hope that the proximity of the table to the ERT discussion will help the reader to follow our arguments more easily.*

*We will change the headline of Table 2: [Resistivity values for different materials derived by field measurement. The used terms of the interpreted material followed Hauck and Kneisel (2008) and Mewes et al. (2017).]*

3c. The authors first reported low InSAR coherence in spring and summer (line 239), but later stated as in spring and autumn (line 270). Did I miss anything? And could the authors add some description of the intermittent SBAS approach? Could you also include velocity ranges in Table 3?

*Thank you for pointing out this inconsistency, it should read spring and summer in both cases. We will add a few sentences to explain the intermittent SBAS approach more clearly and we will adapt Table 3 to include velocity ranges.*

Figure 1: I cannot find blue, red, or black arrows in 1A. Label "Namco"should be "Nam Co"

*Figure 1 will be adapted according the suggestions by the reviewer. By converting a doc-file in a pdf-file these features had been lost.*

Figure 6: The black rectangle is labeled as Figure 10, but should be Figure 9. The colors for lakes and glaciers are very similar (can set a darker color for lakes). It would be ideal to use an identical set of colors for the landforms in both figures 6 and 9.

*Figure 6 will be adapted according the suggestions by the reviewer.*

Figure 9: (this is a minorvisualization issue). InSAR-measured creeping rates show how as light yellow in some landforms (e.g. the two lowest ramparts and in the middle of some rock glaciers), which are not well distinguishable from the light--yellow elevation mask.

*According to this point and the reviewers's point 2, a visual illustration of a lower permafrost limit was dispensed with in Figure 9 due to the new range of 5300 to 5450 m. Instead, a concluding sentence after the corresponding result section will be inserted. Additionally, a small subchapter in L 403 will be inserted:*

*[4.4 Assessement of the lower permafrost limit of the Qugaqie valley*

*The active status, the altitudinal distribution and validated ice-occurrence by ERT of the periglacial landforms display a range of probable permafrost between 5300 – 5450 m a.s.l. This range includes ice lenses detected by ERT-data as well as all creeping landforms indicating an active status and therefore an existence of ice.]*

Editorial comments:
*All editorial comments will be implemented according the suggestions.*

No need to capitalize permafrost if it is not at the beginning of a sentence.

L17: play should be plays

L19: at should be in

L50: decrease should be decreased

L55: spell out ERT here, at its first appearance in the text

L56:part should be parts

L61 & L69:at the TP should be on the TP

L79: which should be what

L93: cretaceous should be Cretaceous

L96: In should be On

L97: spell out ISM here at its first appearance in the text

L118: 'the'' GLIMSdatabase (can also spell out GLIMS)

L120: then should be than

L128: two-years should be two-year

L152: "derived the from the mean"?mean'?

L190:is 'pronival rampart' interchangeable with protalus rampart? If so, it is better to stick with protalus rampart..
*[An incorrect determination as pronival ramparts -a similar looking landform like protalus rampart- can be minimized…]*

L235: changes should be change

L328: "in terms their interpretation in terms of material characteristics"?

L405: are should be is

L408: regimes should be regime

L409: effect should be effects

L410: an should be and?

L410: cower should be cover

L411: delete one 'at'

L412: cause should be because

L418: method should be methods

L431: delete 'the'' before this

L466: of should be away

L466: routing should be rooting?

L469: occurence should be occurrence

L472: scalled should be scaled (or large-scale)

L473: probabilities should be probability

L478: insight is a noun

L483:- asses should be assess

Cao, B., Zhang, T., Wu, Q., Sheng, Y., Zhao, L. and Zou, D.: Brief communication : Evaluation and inter-comparisons of Qinghai-Tibet Plateau permafrost maps based on a new inventory of field evidence, Cryosphere, 13(February), 511–519, doi:10.5194/tc-13-511-2019, 2019.

Eckerstorfer, M., Eriksen, H. Ø., Rouyet, L., Christiansen, H. H., Lauknes, T. R. and Blikra, L. H.: Comparison of geomorphological field mapping and 2D-InSAR mapping of periglacial landscape activity at Nordnesfjellet, northern Norway, Earth Surf. Process. Landforms, 43(10), 2147–2156, doi:10.1002/esp.4380, 2018.

Kneisel, C. and Kääb, A.: Mountain permafrost dynamics within a recently exposed glacier forefield inferred by a combined geomorphological, geophysical and photogrammetrical approach, Earth Surf. Process. Landforms, 32(12), 1797–1810, doi:10.1002/esp.1488, 2007.

Lewkowicz, A. G., Etzelmüller, B. and Smith, S. L.: Characteristics of discontinuous permafrost based on ground temperature measurements and electrical resistivity tomography, Southern Yukon, Canada, Permafr. Periglac. Process., 22(4), 320–342, doi:10.1002/ppp.703, 2011.

Ran, Y., Li, X., Cheng, G., Zhang, T., Wu, Q., Jin, H. and Jin, R.: Distribution of Permafrost in China: An Overview of Existing Permafrost Maps, Permafr. Periglac. Process., 23(4), 322–333, doi:10.1002/ppp.1756, 2012.

Reinosch, E., Buckel, J., Dong, J., Gerke, M., Baade, J. and Riedel, B.: InSAR time series analysis of seasonal surface displacement dynamics on the Tibetan Plateau, Cryosph., 14, 1633–1650, doi:10.5194/tc-2019-262, 2020.

Strozzi, T., Kääb, A. and Frauenfelder, R.: Detecting and quantifying mountain permafrost creep from in situ inventory, space-borne radar interferometry and airborne digital photogrammetry, Int. J. Remote Sens., 25(15), 2919–2931, doi:10.1080/0143116042000192330, 2004.

Sun, Z., Zhao, L., Hu, G., Qiao, Y., Du, E., Zou, D. and Xie, C.: Modeling permafrost changes on the Qinghai–Tibetan plateau from 1966 to 2100: A case study from two boreholes along the Qinghai–Tibet engineering corridor, Permafr. Periglac. Process., 31(1), 156–171, doi:10.1002/ppp.2022, 2020.

---

## Author Comment (AC2) · 13 Aug 2020

This document includes the authors's comments *(italic font)* to the review (normal font) RC2 of Anonymous Referee #2, from 2 July 2020. Planned changes in the revised document are mentioned in […].

Review of "Insights in a remote cryosphere: A multi method approach to assess permafrost occurrence at the Qugaqie basin, western Nyainqêntanglha Range, Tibetan Plateau" by Johannes Buckel et al..
In this interesting work, the authors combined three distinct methods: (i) geomorphological mapping to identify periglacial landforms, (ii) geophysical (ERT) survey to evaluate the presence of ice, and (iii) satellite Synthetic Aperture Radar Interferometry to investigate surface displacements of periglacial landforms that can be used as a proxy of ice occurrence. These methods were applied in the remote Qugaqie basin in Tibet. Despite the logistical problems due to high altitude and extreme climatic conditions, field campaigns were conducted in this area for ERT survey. This study increases the knowledge of poorly studied periglacial geomorphology in Tibet, including the permafrost occurrence within a basin where no borehole measurements are available.

Here some general comments about the manuscript, followed by some detailed comments. A careful proofread is needed.

General comments.
The study area and the three methods used in this work are well described in their respective sections. However, further information on the approaches and criteria used to outline the identified landforms should be added, including some examples with their associated uncertainties.
*Figure 9 will be extended by a new part (A). This part shows a Landsat image and indicates quite well the outlines (represented in very thin lines) of the periglacial landforms. So the reader has the opportunity to see the landform and the associated outline to check the mapping accuracy. The outlines base only on the mapping procedure described in section 3.1, not on InSAR-data. This section will be extended by more information on the used optical images (same issue mentioned by Referee#1).*

About the method that describes the Synthetic Aperture Radar Interferometry, different geometries (i.e. ascending and descending) are used. However, the Authors should explain how they used these different geometries (combination of both, choosing the most appropriate, etc.).
*L250 will be changed to: [Both ascending (satellite travelling south to north) and descending (satellite travelling north to south) acquisitions are therefore sensitive to vertical surface displacement and towards the East or West respectively but very insensitive to displacement towards the North or South. We always select the geometry with the highest sensitivity towards the expected displacement direction to calculate our displacement and velocity results. To that end we calculate a sensitivity coefficient for each pixel which is explained in the following.]*
*The following will be added to L262: [If both ascending and descending velocity LOS data is available for the same pixel, then we use only the geometry with the higher sensitivity coefficient, i.e. better sensitivity, to calculate the downslope velocity and ignore the other geometry to keep the precision of the projection as high as possible.]*

In the results section, velocity values are showed in the text, tables and figures. However, disagreements are apparently visible, probably due to the different "observation periods" in which the velocities were computed (e.g. velocity computed in annual periods vs velocity computed in summer periods?) not specified in the text. The Authors should clarify how the velocity are computed and in which observation periods, in order to avoid misunderstandings.
*These disagreements will be corrected in tables and the text. They did not describe different observation periods (all velocity values in the manuscript refer to the entire observation period of 2015 - 2018) but described different parts*

*of the landforms. We made changes to the tables and text to describe the same parts of the landforms in both. We also added the observation period to a number of figure captions and paragraphs in the text to make it easier for the reader to follow. More detailed information regarding the changes we made can be found under the specific comments (L376 (table 3) + L365) made by the reviewer detailed comment section .*

Furthermore, measurements are often shown with different units, but the same unit should be used (mm/yr and mm or cm/yr and cm).
*This will be checked and all units will be adapted.*

In the discussions there is not a paragraph or section that discusses the mapping of landforms. The same landform can be outlined following different approaches. As suggested by Brardinoni et al. 2019 (https://doi.org/10.1002/esp.4674) the uncertainty on mapping rock glacier is very high. IPA Action Group is now working on baseline concepts and practical guidelines to produce uniform rock glacier inventories (https://www3.unifr.ch/geo/geomorphology/en/research/ipa-action-group-rock-glacier/).

*We follow the geomorphological approach and mapped the extended geomorphological footprint, according baseline concepts. Following the suggestions of reviewer No. 1 and No. 2, we will add more details on the mapping procedure in the Methods section. The outlining of landforms can be subjective, as shown by Brardinoni et al. 2019. However, in our case we are not aiming at quantitative estimates, such as the total area, but only in the existence of certain landforms, to produce an inventory of altitude-zonal localization and to assess the probable occurrence of permafrost.*
*We will add a brief discussion of this aspect in the discussion section.*

Detailed comments
L 56: unclear sentence.
*Will be changed.*

L 78: this sentence seems to be a "general research question". As you already mention, Barsch, 1996; Schrott, 1996, Scapozza, 2015 say this. Specify that this sentence is related to TB landforms.
*We assume that the reviewer means "landforms on the Tibetan Plateau". Because almost no literature was found about ice content of periglacial landforms at the Tibetan Plateau, we ask this general question. We will add a sentence in L 74:[ Especially on the TP only less literature (Fort and van Vliet-Lanoe, 2007; Wang and French, 1995) is available that describes periglacial landforms and permafrost occurrence. However, this periglacial indicators are essential creating large-scale permafrost distribution maps (e.g. Schmid et al., 2015).*

L 97: specify what the ISM acronym is (now it is in the L 106).L 114, Figure 1A: wind systems are not visible in the figure A. Add the position of Nyainqêntanglha in figure A.
*Will be changed.*

L 152 – 154: unclear sentence.
*Will be changed: [So, permafrost occurrence is indicated by activity of landforms and the corresponding surface structures like bulges, furrows, ridges or lobes]*

L 168: here you say "excluding glacially-conditioned environments", but in the following parts you mapped also glaciers. Be clearer.
*"excluding glacially-conditioned environments" will be deleted. The headline of 3.1 will changed to "Inventory of meso-scaled landforms", which includes streams, glaciers, rockglaciers and so on.*

L 171: the meaning of "manually" is unclear (after the field campaign?). Be clearer.
*"Manually mapped information" will be changed to "field-mapped landforms"*

L 173: add the parameters used to derive the hillshade (azimuth and inclination of light source).
*Will be changed.*

L 175: IPA Action Group is working on Baselines and Practical Guidelines for mapping rock glaciers.
*Will be changed: [We follow the geomorphological mapping approach based on the baseline concepts (V 4.0) of the IPA Action Group "Rock glacier inventories and kinematics" and mapped the extended geomorphological footprint of the rockglaciers.]*

L 176: unclear sentence.
*Will be deleted.*

L 181: Location of the photos at Figure 6 and 9.
*Will be added.*

L 236: specify that displacements higher than half of the SAR wavelength are undetectable (e.g. rockfall) and coherence also decreases.
*We will add: [Coherence also decreases with increasing displacement and displacements larger than half the SAR wavelength (~2.8 cm for Sentinel-1) cannot be determined accurately.]*

L 237: How do you choose the threshold of 0.3? Motivate your choice.
*We will add the following sentence to L 238: [This threshold is similar to the one chosen by Sowter et al. (2013) and provides good spatial data coverage while also excluding unreliable data.]*

L 254: Describe how you used the ascending and descending dataset (do you combine them? Or do you select one depending on the landform orientation?).
*This issue concerns to a previous comment of the reviewer. Accordung to that we will change L250 to: [Both ascending (satellite travelling south to north) and descending (satellite travelling north to south) acquisitions are therefore sensitive to vertical surface displacement and towards the East or West respectively but very insensitive to displacement towards the North or South. We always select the geometry with the highest sensitivity towards the expected displacement direction to calculate our displacement and velocity results. To that end we calculate a sensitivity coefficient for each pixel which is explained in the following.]*
*The following will be added to L262: [If both ascending and descending velocity LOS data is available for the same pixel, then we use only the geometry with the higher sensitivity coefficient, i.e. better sensitivity, to calculate the creeping rate and ignore the other geometry to keep the precision of the projection as high as possible.]*

L 262: coefficient lower than 0.2 is possible, but it is not used in order to avoid an excessive amplification of displacements and associated errors (from Notti et al. 2014).
*We will change L 262 to: [The coefficient can vary between 0 for areas where the satellite's sensitivity is low to 1 where the sensitivity is very high. Values below 0.2 are not used to avoid excessive amplification of displacements and associated errors.]*

L 274: a table containing the parameters used during the processing (or the reference) can be useful here.
*We will add the following table:[Table 1:Summary of ISBAS processing parameters.]*

| Geometry | Observation period | Acquisitions | Interferograms | Temporal baseline | Coherence threshold |
|---|---|---|---|---|---|
| ascending | 2015-06-05 to | 74 | 278 | 12 to 72 | 0.3 |

| | 2018-12-22 | | | days | |
|---|---|---|---|---|---|
| descending | 2015-11-15 to 2018-12-29 | 63 | 257 | 12 to 96 days | 0.3 |

L 277: provide the locations and outlines of these stable areas (e.g. in figure 6 or 9).
*Will be added to Figure 9.*

*L278 will be changed to: [Reference points are located on bedrock whenever possible and on ridges or moraines with good coherence if no coherent bedrock was available.]*

L 300, figure 6: Inside the figure 6 replace Figure 10 with 9.
*Will be changed*

L 312: talus slope and protalus rampart also provide sediments in the rooting zone of rock glaciers (visible in figure 6).
*Will be changed: [Moraine deposits, talus slopes and protalus ramparts provide…]*

L 346, figure 8: use the "k ohm m" unit in the figure 8 (as in the text). Add the cartesian coordinates on each profile (e.g. West and East for A A').
*Will be changed*

L 365: specify how you compute the mean velocity of the landforms, not only in the caption of table 3 (do you selected some points inside the landforms? Or do you use all the points inside the delineated landforms?), and add the observation periods (e.g. velocity computed in annual periods or velocity computed summer periods?).
*To specify this issue we will change L252 to: [Mean velocities were calculated by dividing the cumulative displacement observed during the observation period by the length of the observation period (2015-2018). All surface velocity data of periglacial landforms have been projected along the direction of the steepest slope under the assumption that the motion of the described landforms is mainly gravity-driven. Hereafter we will refer to the mean surface velocity of periglacial landforms projected along the steepest slope as "creeping rates" to reflect this assumption.]*

*L366 will be changed to: [Mean creeping rates during the entire observation period (2015-2018) for all data points within the landforms are higher on rockglaciers (26.8 mm/yr) than protalus ramparts (12.7 mm/yr).]*

*In general, we write creepings rates if we mean displacement in slope direction. This will be added in the Chapter 3.3, too*

L 371: use always the same units in the text (mm/yr and mm or cm/yr and cm).
*Will be changed*

L 376, table 3:
- the computation of "downslope velocity precision" is not clear.
- the "interpolated time periods" is not defined.
- there are apparent disagreements between this table and the text below. For example, from this table the maximum downslope velocity expected for rock glacier is 47.2 mm/yr (26.8±20.4), but velocities up to 153 mm/yr are visible in figure 9 and 10. This is probably due to different points

investigated or different "observation periods" where the mean velocities were computed (e.g. annual periods vs summer periods?). please specify how these values are obtained.
*Table 3 displays the mean downslope velocity and the standard deviation, while the values shown in figures 9 and 10 represent the range of the downslope velocity. We will add a column in Table 3 describing the range of rockglaciers and protalus ramparts to make this clear. We will add the following to the caption of Table 3: [Creeping rate precision is calculated by dividing the LOS precision of 2.4 mm/yr by the sensitivity coefficient. The percentage of interpolated time periods describes how many interferograms are incoherent and therefore require interpolation with the ISBAS algorithm.]*

L 386, Figure 9: Specify if the "creeping rates" are along LOS or slope direction. Indicate the observation period of the velocities showed in the figure.
*The creeping rates are along the slope direction and the observation period is from 2015 – 2018. This will be added in the figure caption.*

L 390: these values ("from 10 mm/yr to 80 mm/yr") are apparently in disagreement with those in table 3 (the mean downslope velocity is 12.7±9.2 mm/yr). Explain why (different observation periods?).

*The values used in the text describe only a part of the data points, while the values in the table describe all data points on the respective landforms. This should have been mentioned in the text. As this is confusing to the reader and unnecessary, we decided to use the same values in the table and the text. In L366 we mention the mean velocities of the landforms. It should therefore not be necessary to mention them again in L386. L386 will therefore be changed to: [The creeping rate of protalus ramparts is lower and shows more pronounced seasonal variations than that of rockglaciers.]*

L 397 – 398: this sentence is not supported by results and data. Time series of temperature and precipitation are not shown. Provide data (or references) to support, or remove this sentence.
*We will remove the sentence*

L 401, figure 10: indicate the observation periods of the velocities showed in the maps B and C.
*The figure caption will be adapted. Observation periods were mentioned for Map A, B and C.*

L 433: specifies whether the layer refers to the ice layer.
*It refers to the active layer thickness. "active" will be added*

L 466 – 469: rephrase these two sentences (it is not clear that these two sentences are not related to your work).
*These lines will be changed: [Using rockglaciers and their long-term ice content as indicators for permafrost occurrence must be critically evaluated because rockglaciers can overcome long distances and the terminus is far away from the routing zone (Bolch and Gorbunov, 2014; Halla et al., 2020)]*

L 483: Sentinel 1 is not a high-resolution satellite.
*We will delete "high–resolution".*

---

## Author Response (AR1)

Dear Editor,

Thank you for the information about "publishing subject to minor revisions". This response letter includes three answer-sections to the main comments of the editor, the reviewer 1 and the reviewer 2. Editorial comments regarding grammatical and language issues are not mentioned here, but are carefully inserted in the manuscript. Here, we address the main comments and how we answered to them.

Attached at this letter to the editor you will find the manuscript with tracked changes.

Answer to Editors comments:

Nature as a continuum couldn't be classified by clear boundaries, especially at our study-scale. So, we changed the main aim from the study to give a prediction of the lower permafrost (PF) limit to an assessment of the lower occurrence of probable permafrost. This change can be tracked through the manuscript by different sections. We put the assessment of the lower occurrence of probable PF in a better context of existing data (e.g. MAAT of the weather station of Zhadang). Two additional paragraphs regarding the atmospheric and topographic influence on PF-occurrence discuss the results of the PF-assessment in the discussion section with corresponding literature. Rockglaciers as permafrost indication can be used but carefully as the editor stated out. Therefore, we discussed corresponding literature and stated out that the rockglaciers in our study area are suitable for a zonal permafrost assessment. The reason for this lies in the low elevational range of these landforms. The distance between the rooting zone and the terminus is short both in length and elevation in comparison to other rockglaciers. So, the rockglaciers do not creep out the periglacial environment.

The uncertainty for displacement measurements were given by a sensitivity coefficient included by the co-author E. Reinosch based on his publication (https://doi.org/10.5194/tc-14-1633-2020).

Instead of a zoom in the permafrost map(s) for the investigated catchment in the Fig. 1, we add in the Fig. description: Note the proposed Permafrost occurrence of more than 90 % in study area Qugaqie Basin (black feature).

Answer to Reviewer 1 comments:

We agree that the term "probable permafrost" is not well defined, and that the lack of a precise definition might cause confusion. Therefore, we change the term to "probable occurrence of permafrost". The use of the term "probable" is motivated by the fact that we do not have borehole-data (MAGST) and a modelled permafrost distribution and we can only assess the occurrence indirectly. The spatial heterogeneity of our data (mapping, InSar and ERT) and of natural variations in permafrost occurence also prevents us from providing precise limits. This issue goes strongly in the direction to the editor's comment (see above).

We answer the reviewer's remaining question in the quantitative interpretation of the elevation (or elevation ranges) as inferred from each method in the final manuscript as follows. Each result section of the corresponding methods received a concluding sentence indicating the inferred elevation of probable permafrost occurrence.

One aspect of our study is that it provides localized ground-truth data in an area that is otherwise only accessible by simulation or theoretical considerations within large-scale models. Therefore, in order to compare the result with existing literature, we find it useful to infer one single value. In reality, the occurrence of permafrost naturally varies depending on topo-climatic factors (see editors comments). To take this into account, the single value is transformed into a range, which also includes the two lowest, active protalus ramparts. We state out as a general outcome of the publication "the probable occurrence permafrost is higher than 5300 - 5450 m. a.s.l"

The added values of the ERT-results, are that ERT supplements InSar-findings and vice versa. Surface creep without ice could happen without permafrost conditions and subsurface ice without creep over 2 consecutive years is not related to permafrost. With the help of ERT we were able to provide evidence for the existence of ground ice at specific test sites. The localized site surveys are critical for arriving at the 5300 – 5450 m estimate. Profile A (Fig. 8) ranges from 5090 to 5230m and no subsurface ice content was detected in the lateral moraine, which could be supposed.

Due to the reviewers aspect of a marginally supplement of InSAR-data, we believe that the surface movement, derived by InSAR, is one key parameter of periglacial landscapes (Eckerstorfer et al., 2018) and is used to infer permafrost conditions (Kneisel and Kääb, 2007; Strozzi et al., 2004). Therefore, InSar-data plays a major role in this study, because this data validates the two-years rule of frozen ground (=permafrost) through the creeping behavior (compare Fig. 10A).

A differentiation of seasonal and perennial frozen ground of the study area is given by Reinosch et al. (2020). Based on InSar time series analyses of the typical periglacial surface changes the authors developed a model to distinguish annual surface changes (seasonal frozen ground) from perennial, more or less constant creeping surface changes (perennial frozen ground) for the study area. We focus directly on periglacial landforms according to perennial creep caused by down moving ice content.

Regarding moving the Table 2 and the corresponding text to section 3.2. is probably debatable but since the compiled resistivity values are results and are needed for the explanation of the ERT results, we would prefer to leave the table in section 4.2. We hope that the proximity of the table to the ERT discussion will help the reader to follow our arguments more easily. So, we will change the headline of Table 2: Resistivity values for different materials derived by field measurement. The used terms of the interpreted material followed Hauck and Kneisel (2008) and Mewes et al. (2017).

More details about the mapping procedure were mentioned in the method section.

Answer to Reviewer 2 comments:

As the reviewer 2 mentioned different geometries (i.e.ascending and descending) are used in the description of the Synthetic Aperture Radar Interferometry. We explained the use of different geometry in the corresponding method section. Both ascending (satellite travelling south to north) and descending (satellite travelling north to south) acquisitions are therefore sensitive to vertical surface displacement and towards the East or West respectively but very insensitive to displacement towards the North or

South. We always select the geometry with the highest sensitivity towards the expected displacement direction to calculate our displacement and velocity results. To that end we calculate a sensitivity coefficient for each pixel which is explained in the following. If both ascending and descending velocity LOS data is available for the same pixel, then we use only the geometry with the higher sensitivity coefficient, i.e. better sensitivity, to calculate the downslope velocity and ignore the other geometry to keep the precision of the projection as high as possible.

All observation periods of the shown surface velocity refer to the entire observation period of 2015 - 2018 but described different parts of the landforms. We made changes to the tables and text to describe the same parts of the landforms in both. We also added the observation period to a number of figure captions and paragraphs in the text to make it easier for the reader to follow.

We changed all units to mm/y for an easier comparability.

Now, the discussion chapter describes now a section about the uncertainty of the mapping procedure of the periglacial landforms which contains the IPA mapping guidelines and the important literature by Brardinoni et al. (2019).

Literature:

[revised manuscript text omitted]

---

## Author Response (AR2)

Dear Editor,

this document contains a point by point reply to your comments and suggestions. The editorial comments are in black and author's comments in blue. Line numbers refer to the document where changes are visible attached at this point by point reply (behind page 11).

L23ff: This new sentence is now very long and difficult to understand. Split into two and clearly state that your result indicates a lower limit of permafrost between 5350 and 5500 m asl. (or 5450 m asl. as you started in the response letter?)

> 5450 m a.s.l. was a mistake because the lowest occurrence of a protalus rampart defines the lower boundary of 5350 m a.s.l. We split and re-structure the sentence as follows:
>
> L23ff: A multi-method-approach combines (A) geomorphological mapping, (B) electrical resistivity tomography (ERT) to identify subsurface ice-occurrence, and (C) Interferometric Synthetic Aperture Radar (InSAR) analysis to derive multi-annual creeping rates. The combination of the resulting data allows an assessment of the lower occurrence of permafrost in a range of 5350 and 5500 m above sea level (a.s.l.) in the Qugaqie basin.

L26: Please write a.s.l. in full the first time mentioned.

> This is now included in the change in the comment above

L29f: "However, seasonality…". Sounds like you missed this in your study as well. Either omit or move to the beginning of the abstract and clearly mention the results about the seasonality of your study.

> We decided to omit the topic of seasonality in the abstract. New data about this special rockglacier show that the lack of seasonality is due to a transitional state from a glacial to a periglacial landform. This aspect will be subject of another publication.

L31f: Which other mountain areas? Are there regions with higher permafrost occurrence?

> We delete the second part of the sentence due to the length of the sentence and to a lesser importance of the comparison in the abstract.

L35: Include here the later mentioned thermal definition with the 2 consecutive years.

> L39:(defined as a thermal state of perennially cryotic ground, at least frozen of two consecutive years (Ballantyne, 2018; Washburn, 1979))

L37: Provide also a reference which focusses specifically on permafrost and not only on rock glaciers.

The article of Coline Mollaret is cited in L42: https://doi.org/10.5194/tc-13-2557-2019

L38: "Due to its thermal definition, it is difficult to evaluate permafrost conditions without ice-occurrence." This sentence is unclear. Can't you measure temperatures to evaluate permafrost occurrence? Please revise.

Yes, with borehole temperature loggers it is possible to measure permafrost conditions. But the drilling in remote environments is rather difficult and often impossible. Since the discussion of this challenge is not appropriate at this point in the introduction we decided to skip the sentence.

L40: I disagree that "Periglacial landforms of high mountain environments well suited to detect and to study changes of permafrost temperature." They might be suited to detect permafrost conditions and changes but not of the temperature. "A periglacial landform is a feature resulting from the action of intense frost, often combined (rem.: but not necessarily) with the presence of permafrost." (French 2012). Moreover: State here or where it fits what landforms you are investigating.

Here, we skip "temperature" *(It was orginally included to state permafrost out again as a thermal phenomena)* and separate the sentences by including the definition by French 2012 and the investigated landforms of this study:

L45: Periglacial landforms, like rockglaciers and protalus ramparts in this study, are features "resulting from the action of intense frost, often combined with the presence of permafrost" (French, 2012). If permafrost as perennial frozen ground ice is available, periglacial landforms are particularly well suited to detect and to study changes of permafrost and the related ice content (Kneisel and Kääb, 2007, Kääb, 2013, Knight et al., 2019).

Here or elsewhere in the introduction/discussion I recommend also to consider also other holistic studies combining filed investigations and remote sensing to investigate rock glaciers, their ice content and climatic response (e.g. Monnier et al. 2014, Dusik et al. 2015, Bosson and Lambiel, 2016; Bolch et al., 2019)

L93: The combination of field investigations and remote sensing techniques is a useful tool to detect permafrost occurrence (Bolch et al., 2019; Dusik et al., 2015; Monnier et al., 2014)

L43: Refrain from citing submitted papers (I thought Chris Halla's manuscript will be accepted until the end of the review). There are several studies addressing the importance of rock glaciers in arid South America (e.g. Rangecroft et al. 2016, Azocar and Brenning 2010). I suggest also to include references for HMA.

L51: of long-term ground ice as water resource (Jones et al., 2019) in arid/semiarid regions like the Andes (Azócar and Brenning, 2010; Rangecroft et al., 2016) or the Tian Shan (Bolch and Gorbunov, 2014; Bolch and Marchenko, 2006).

L45: Please update with Immerzeel et al. (2020).

L55: New citation http://dx.doi.org/10.1038/s41586-019-1822-y

L48: Please use "atmospheric warming". Is there any more recent reference in addition?

L58: climate is replaced by atmospheric. Citation of 10.1038/s41598-017-04140-7 was inserted (Lu et al., 2017)

L56: Chen et al. (2016) investigate an area on the central-west TP. It is a nice study but not really suited to back up this statement. I suggest cite a review paper instead, which provides an overview of available measurements.

L67: Chen et al 2016 is replaced by the review of Yang et al. (2010)

L69: The study area is certainly remote, but, e.g. the railway line is – considering the vast area of Tibet relatively close.

L80: We replaced "away from the Tibetan engineering corridors" by "unbiased by the location of the Tibetan corridors" in order to remove the aspect of spatial distance.

L71: "MAGST". It is not clear to me. You do not need boreholes to measure the MAGST.

L82: We replace MAGST by "ground truthed temperature data for geophysical data validation"

L72: Write consistently "InSAR"

L85: It is replaced and checked for the entire document.

L83: Here ore elsewhere: Please consider the study by Heaberli et al. (2006), which provides a comprehensive review and includes many co-authors.

In Line 127.

L84f. There is now little more literature presenting rock glaciers on the TP (e.g. Ran et al. 2018). Please be also clear what you mean with periglacial landforms/indicators.

L99ff: Ran et al 2018 is added and the following sentence is rewritten: However, these periglacial landforms as indicator for permafrost occurrence are essential for creating large-scale permafrost distribution maps (e.g. Schmid et al., 2015).

L89f: State which periglacial landforms you are investigating; rock glaciers contain ice per definition unless they are relict. Adjust/revise this research question.

L104ff: Research questions were adjusted:

- Do the investigated periglacial landforms like rockglaciers and protalus ramparts show an active status?
- Which creeping rates do the periglacial landforms indicate?

L98: As stated in my access review you are investigating (and also showing in Fig. 1) Western Nyaingentanglha. Please correct.

"Western" was added in L 115, 122, 123, 126, 129, 149, 364, 617

L110: Add at least one further reference about glacier changes (especially mass changes), e.g. Zhang and Zhang (2017).

In L 126 we added: Zhang and Zhang (2017) observe a melting rate −0.30 ± 0.07 m yr$^{-1}$ over the entire western Nyainqêntaglha range from 2000 to 2014.

Figure 1: I suggest to include a hillshade in Fig. 1B and C. Not clear why there are not data. Please mention in the main text. The sentence "Note the proposed permafrost coverage of more than 90 % in the study area, marked by the black edging." is unclear.

The background of Map B was hillshaded.  By hillshading of Map C a kind of overload occuredshould be clear as possible because otherwise it would be overloaded. The figure caption was adapted by including: (Hillshade and DEM background based on SRTM DEM v4; Jarvis et al., 2008).

The sentence "Note…" was deleted. Instead in the main text L157 we added: "…and covers more than 90% of the study area. The visible data gaps were not further discussed by (Zou et al., 2017)"

"Not clear why there are not data."

We agree that this is a bit unsatisfactory. We tried to find a reason for the data gaps in the original data, without success. Since the gaps are not critical for the conclusions of our paper, we hope they are acceptable.

Figure 2: The arrows are not clear. Why are there no arrows from the method to the aim?

The figure was adapted:

[Figure]

L191: Basically fine as you do not focus on glaciers, but why did you not use the glims outlines as basis?

The Glims outlines are from 2009. In this publication the most recent outlines are presented (2013).

L192f: Please be more specific and mention here how you identified rock glaciers instead of referring only to literature.

L214: Barsch (1996): If the form shows a tongue- or a lobate shape in the field and at the optical images, we classified the landform as a rockglacier. Additionally, field observations like coarse clasts at the surface and at the front indicate typical rockglacier substrate. Protalus rampart are classified by a coarse debris accumulation in front of a rock wall. A small depression occurs between the non-lobate buldge and the weathering rockwall.

Cite the IPA action group report properly already here.

L219: (Delaloye et al., 2018; Delaloye and Echelard, 2020)

L280ff: In contrast, here you describe much detail. Is this section basically a description of the work by Reinosch et al. (2020)? If yes, please shorten keep only the most important information and refer to the paper for more details.

L304 was deleted.

L307/308 were moved to L328

L308-315 were moved to L338 behind Table 1

L315-322 were deleted.

L330-335 were deleted

Figures 6/7/9: The colour chosen are a bit misleading: Dark green give the impression about vegetation. Please choose other colours. E.g. dark brown can also be well distinguished. From the other colours. Moreover the different greenish colours are a bit difficult to distinguish in Fog. 9.

Colours are changed.

L351: I am not fully sure if you do: Please make sure to relate these geomorphological evidences with the remote-sensing derived measurements. Maybe you can even refer in this section to the measurements.

L386: Extended Sentence: "These field observations in combination with the observed creeping rates (Figure 9, B) allow the conclusion of an active status of the rockglaciers"

Caption/Table 4: You do not need to repeat methods in the figure caption but make sure that the info is given in the methods section.

L 351ff contains the info of the caption.

L464: A mistake was detected and changed: the figure 9A is based on Sentinel II satellite, not on Landsat 8.

But even more important: It is not clear where the standard deviation is shown, e.g 13 +- 60? That is a huge spread then. How is the distribution? Normal or skewed?

It is skewed distribution, therefore we now give the median and the interquartile range as uncertainty. According to this, all values in the document were adapted and the median is shown. The new table and caption:

**Table 1: Summary of InSAR- derived creeping rates for the periglacial landforms. The values represent the median of all data points over the entire observation period (2015-2018) on the respective landform. Uncertainty is given by the interquartile range in round brackets. The percentage of interpolated time periods describes how many interferograms are incoherent and therefore require interpolation with the ISBAS algorithm.**

| Landform | Creeping rate [mm/yr] | Summer acceleration [%] | Creeping rate precision [mm/yr] | Coherence | Interpolated time period [%] | Data points |
|---|---|---|---|---|---|---|
| Protalus ramparts | 11.0 (6.8 to 16.7) | -2 (-36 to 36) | 5.2 (3.9 to 7.8) | 0.74 (0.70 to 0.80) | 2.5 (0.7 to 6.2) | 7984 |
| Rockglaciers | 21.1 (11.6 to 36.8) | -23 (-46 to 3) | 5.1 (4.2 to 8.1) | 0.66 (0.59 to 0.73) | 7.9 (3.8 to 13.0) | 5402 |

L432: Add uncertainty.

The uncertainty is given in the new Table 4 and the values of L469/470 were changed to the median and extended by the interquartile range in ().

L568, L607, L608: Mean was changed to median.

Fi. 10: Include a legend for 10A. In this way the figure is understandable right away and you do not write long explanations in the caption.

A legend is included and L 481 and 484are deleted.

The time series points are a bit hard to identify especially in Fig. 10C. I suggest to highlight with an arrow in addition.

Arrows were inserted.

Section 4.4.: This section is a bit short. Put more emphasis on it and provide more information.

L490ff were extended:

The assessment of the lower permafrost limit consists of an integration of different results. The field based mapping of periglacial landforms indicates the first precondition to find permafrost conditions. Field observations like furrows, ridges, coarse substrate and lichen coverage on the rockglaciers surface corroborate the mapped landforms classification and indicate activity of the landform. By Integrating the ERT-results of detected subsurface ice occurrence, a further component of the permafrost condition (subsurface below 0 degree) is validated. Completing the permafrost definition (of two or more consecutive years) the derived creeping rates by InSAR show a constant motion of more than two years which is attributed to the deformation of the debris ice matrix of the periglacial landforms. So,...

L461ff: Harris and Pederson (1998) are not correctly referred to. They e.g. write "Thermal response to a change in air temperature (positive or negative) is immediate and substantial, so it is not merely the result of the Balch effect." Please differentiate better the different processes involved and adjust your text. Please consider to add more literature, e.g. from on rock glaciers in the Alps with detailed measurements. I also recommend to include Gorbunov et al. (2004).

We changed L503ff:

These landforms are characterized by blocky material and a special thermal regime that lowers the internal temperature in comparison to the thermal regime outside of the blocky, rough surface (Gorbunov et al., 2004). This cooling effect of high-porosity unconsolidated debris is especially observed in lower mountain regions by near-surface ground temperature measurements on rockglaciers (Onaca et al., 2020) and suggests a lowering of discontinuous and sporadic permafrost

occurrence (Lambiel and Pieracci, 2008; Otto et al., 2012). By using the ERT-method we found ice-poor permafrost in ice-lenses in mineral soils next to the rockglacier that corroborates the idea of permafrost conditions outside of blocky material at an elevation of 5450 m a.s.l. The extreme cold mean annual air temperature of -6.8°C at 5680 m a.s.l (Zhang et al., 2013) should minimize the effect of different regolith properties that favors permafrost conditions.

L462: "Alps" is no longer in the manuscript, because of the rewritten section L503ff.

L494 "Remote sensing" is changed.

L504ff: Moisture availability is key for the variations in creep and I agree therefore with your argumentation in L 513. Please consider more recent literature where the importance of water for the velocity variations are investigated more in detail (e.g. Wirz et al. 2016; Kenner et al. 2017; Cicoira et al. 2019).

L556: Cicoira et al. 2019 was added

L559: The lack of seasonality and the lower creeping rates compared to rockglaciers in the alps (Cicoira et al., 2019; Kenner et al., 2017; Wirz et al., 2016)…

L520: These references do not fully fit. Please consider at least two with really address the hydrological importance and their response to climate

→L572: References were added and the text slightly modified:

…indicators of stored water resources (Azócar and Brenning, 2010; Jones et al., 2018b, 2018a) and their response to climate (Cicoira et al., 2019; Humlum, 1998).

L574: Reference was added (Delaloye et al., 2018)

L577: Reference was added (Delaloye and Echelard, 2020)

One remark to the author contributions and authors: According to the Copernicus policy (and also to international standard) "All authors listed on a presented scientific work must have contributed a significant part to it. Vice versa, all persons who contributed to the presented work need to be named in the list of authors." You list 5 persons responsible for funding acquisition but only one of them seemed to have really significantly contributed to the work according to the author contribution statement. Please be more specific about the authors' roles and consider revisiting the authors' list.

The section of team list and Author contributions was adapted:

[revised manuscript text omitted]

---

## Author Response (AR3)

Dear Editor!

Thank you very much for accepting the manuscript in The Cryosphere after your suggested corrections.

We omit Bolch & Gorbunov (2014) L44 and we changed L604 to:

Bolch, T., Marchenko, S. S. (2009): Significance of glaciers, rockglaciers and ice-rich permafrost in the Northern Tien Shan as water towers under climate change conditions. In: L. Braun, W. Hagg, I. V. Severskiy & G. J. Young (Eds.): Selected papers from the Workshop "Assessment of Snow, Glacier and Water Resources in Asia" held in Almaty, Kazakhstan, 28-30 Nov. 2006, IHP/HWRP-Berichte, 8: 132-144.

I am very grateful to you, Tobias Bolch, that you have taken so much effort to improve the study by your suggestions.

Johannes Buckel